# Male pheromones modulate synaptic transmission at the *C. elegans* neuromuscular junction in a sexually dimorphic manner

Kang-Ying Qian[1,2,3], Wan-Xin Zeng[1,2,3], Yue Hao[1,2,3], Xian-Ting Zeng[1], Haowen Liu[4], Lei Li[4], Lili Chen[5], Fu-min Tian[1,2], Cindy Chang[6,7], Qi Hall[6,7], Chun-Xue Song[8,9], Shangbang Gao[5], Zhitao Hu[4], Joshua M Kaplan[6,7], Qian Li[8,9,10]\*, Xia-Jing Tong[1]\*

[1]School of Life Science and Technology, ShanghaiTech University, Shanghai, China; [2]University of Chinese Academy of Sciences, Beijing, China; [3]Institute of Neuroscience, Shanghai Institutes for Biological Sciences, Chinese Academy of Sciences, Shanghai, China; [4]Queensland Brain Institute, Clem Jones Centre for Ageing Dementia Research (CJCADR), The University of Queensland, Brisbane, Australia; [5]College of Life Science and Technology, Huazhong University of Science and Technology, Wuhan, China; [6]Department of Molecular Biology, Massachusetts General Hospital, Boston, United States; [7]Department of Neurobiology, Harvard Medical School, Boston, United States; [8]Center for Brain Science, Shanghai Children's Medical Center, Shanghai, China; [9]Department of Anatomy and Physiology, Shanghai Jiao Tong University School of Medicine, Shanghai, China; [10]Shanghai Research Center for Brain Science and Brain-Inspired Intelligence, Shanghai, China

\*For correspondence:
liqian@shsmu.edu.cn (QL);
tongxj@shanghaitech.edu.cn (X-JT)

Competing interests: The authors declare that no competing interests exist.

**Abstract** The development of functional synapses in the nervous system is important for animal physiology and behaviors, and its disturbance has been linked with many neurodevelopmental disorders. The synaptic transmission efficacy can be modulated by the environment to accommodate external changes, which is crucial for animal reproduction and survival. However, the underlying plasticity of synaptic transmission remains poorly understood. Here we show that in *Caenorhabditis elegans*, the male environment increases the hermaphrodite cholinergic transmission at the neuromuscular junction (NMJ), which alters hermaphrodites' locomotion velocity and mating efficiency. We identify that the male-specific pheromones mediate this synaptic transmission modulation effect in a developmental stage-dependent manner. Dissection of the sensory circuits reveals that the AWB chemosensory neurons sense those male pheromones and further transduce the information to NMJ using cGMP signaling. Exposure of hermaphrodites to the male pheromones specifically increases the accumulation of presynaptic CaV2 calcium channels and clustering of postsynaptic acetylcholine receptors at cholinergic synapses of NMJ, which potentiates cholinergic synaptic transmission. Thus, our study demonstrates a circuit mechanism for synaptic modulation and behavioral flexibility by sexual dimorphic pheromones.

## Introduction

Faithful synaptic transmission is essential for animal physiology and behaviors. The disturbance of synaptic transmission has been linked with several neurodevelopmental disorders, including autism

spectrum disorders (ASD). In the past decades, researchers have identified numerous genes encoding synaptic proteins that are linked with neurodevelopmental disorders, and their mutations cause the dysregulated synaptic transmission in human diseases (*Homozygosity Mapping Consortium for Autism et al., 2016*; *Autism Sequencing Consortium et al., 2019*; *Geisheker et al., 2017*; *Iossifov et al., 2012*; *Lee et al., 2019*; *Morrow et al., 2008*; *Neale et al., 2012*; *C Yuen et al., 2017*), including *SHANK3, NRXN,* and *NLGN* for autism (*Chen et al., 2020*; *Lee et al., 2015*; *Levinson and El-Husseini, 2005*; *Orefice et al., 2019*; *Südhof, 2008*), *MECP2* for Rett's syndrome (*Chao et al., 2007*; *Orefice et al., 2019*), *FMR1* for Fragile X syndrome (*Olmos-Serrano et al., 2010*), and *UBE3* for Angelman syndrome (*Judson et al., 2016*; *Wallace et al., 2012*).

The process of synaptogenesis occurs in the early postnatal developmental period and can be modulated by the environment. The effects of synaptic modulation could persist until adulthood and cause a lifelong impact. Various environmental contexts can modulate synaptic transmission and behaviors through experience-dependent plasticity, which provides a critical and conserved mechanism to generate animal behavior diversity and adaption. Among the environmental contexts, social interaction, such as the density of the conspecifics sharing the same habitat, represents one of the most important environmental conditions that modulate animal physiology and behaviors to meet the ever-changing environment and internal needs (*Chen and Hong, 2018*). For example, social isolation of rats during the critical period of adolescence enhances long-term potentiation of NMDA receptor-mediated glutamatergic transmission in the ventral tegmental area (*Whitaker et al., 2013*). Besides that, maternal separation has been found to have a profound lifelong influence on animal models at a mature stage of life. It causes habenula hyperexcitability, AMPA receptors delivery, and synaptic plasticity defects in the developing barrel cortex (*Miyazaki et al., 2012*; *Tchenio et al., 2017*). However, the underlying mechanism on how social interaction modulates synaptic transmission remains elusive.

There are many ways in which social interaction can influence neural development. Pheromone effects between conspecifics are strong drivers that modulate behaviors and alter physiology, allowing appropriate responses to particular social environments (*Liberles, 2014*). These effects are often sexually dimorphic. Mouse pups elicit parental care behaviors in virgin females, for instance, but promote infanticidal behaviors in virgin males through pheromonal compounds (submandibular gland protein C and hemoglobins) and physical traits (*Isogai et al., 2018*). In *Caenorhabditis elegans* (*C. elegans*), a family of glycolipids called ascarosides function as the pheromones to mediate social interactions. Males and hermaphrodites secrete several ascarosides in different amounts that elicit sexual dimorphic responses (*Butcher et al., 2007*; *Edison, 2009*; *Greene et al., 2016*; *Srinivasan et al., 2008*; *Srinivasan et al., 2012*). For example, the male-enriched ascr#10 induces attraction behavior in hermaphrodites, but causes aversion behavior in males (*Izrayelit et al., 2012*). However, it remains unclear whether and how specific pheromone-mediated effects are involved in neurodevelopmental processes, including synaptogenesis and synaptic transmission.

Here, we show that the male environment increases the cholinergic synaptic transmission at the neuromuscular junction (NMJ) in *C. elegans* hermaphrodites, decreasing hermaphrodite's locomotion activity and promoting mating efficiency. The male-specific pheromones (ascarosides) mediate these effects in a sexually dimorphic manner. Such ascaroside-mediated modulation of the cholinergic synaptic transmission is developmental stage dependent. We further used various neuron-type-specific ablation experiments to confirm that these male-specific pheromone signals are received and processed by the AWB chemosensory neuron pair in hermaphrodites. Upon reception, AWB neurons transduce the information to the NMJ using cGMP signaling. Furthermore, we used multiple reporter fusion constructs to show that the male-specific pheromones cause increased calcium channel accumulation and acetylcholine receptor (AchR) clustering at cholinergic synapses. Collectively, our work elucidates how individuals sense and adapt to the social environment, providing insights into how pheromones regulate the development and function of the nervous system.

## Results

### The male environment modulates synaptic transmission at the hermaphrodite NMJ

*C. elegans* has two sexes: XX hermaphrodites and XO males. The somatically female hermaphrodites can produce hermaphrodite progeny by self-fertilization (although rare males are generated through spontaneous X chromosome loss), whereas in the presence of males, they are also able to mate with males to give rise to equal ratios of hermaphrodites and males (*Figure 1A*). Hermaphrodites generated by self-fertilization or by crossing share the same genetic background but develop in distinct environments (i.e., in the presence or absence of males). Therefore, it provides an excellent system to study how social interaction modulates the establishment and maintenance of synaptic transmission during development. We selected the *C. elegans* NMJ as a model to examine the male environment's effects on synaptic transmission. The *C. elegans* NMJ includes body-wall muscles that receive synaptic inputs from both excitatory cholinergic and inhibitory GABAergic motor neurons (*Richmond and Jorgensen, 1999*). The coordination of excitatory and inhibitory innervations guarantees *C. elegans* sinusoidal movement. In the presence of acetylcholinesterase inhibitors such as aldicarb, the breakdown of acetylcholine is prevented, and acetylcholine accumulates over time at synapses. As a result, worms become paralyzed due to hyper-excitation (*Mahoney et al., 2006*). The timing of the paralysis is influenced by the inhibitory innervations from GABAergic neurons that counteract acetylcholine's excitatory effect and delay paralysis. The percentage of paralyzed worms over time can be used as a measurement of excitatory versus inhibitory synaptic transmission ratio (E/I ratio) at the NMJ. As a result, the alteration of sensitivity to aldicarb reflects the changes in NMJ synaptic transmission (*Vashlishan et al., 2008*).

To determine whether the NMJ synaptic transmission differs between hermaphrodites generated through self-fertilization versus crossing, we applied aldicarb to young adult hermaphrodites and examined the percentage of paralyzed animals. We found that around 39.8% of hermaphrodites from self-fertilization were paralyzed after 70 min' exposure to aldicarb. In contrast, almost all of the hermaphrodites from crossing were paralyzed (*Figure 1B*), indicating that hermaphrodites obtained by crossing are more sensitive to aldicarb. Thus, the NMJ E/I ratio is increased in crossed hermaphrodites than those obtained by self-fertilization.

There are three possible explanations for the observed differences in NMJ synaptic transmission in crossed hermaphrodites: first, it could be a parental inheritance effect, such as RNA transgenerational transmission (*Alcazar et al., 2008*; *Rechavi et al., 2011*); second, it could be caused by direct contact with males (*Shi and Murphy, 2014*); third, male metabolites secreted into the environment could modulate hermaphrodite development. To rule out the potential effects of parental inheritance and male contact, we directly exposed hermaphrodites from self-fertilization to medium conditioned by either the male or the hermaphrodite environment since egg stage. The conditioned medium was prepared by collecting cultures of *him-5* mutants containing around 40% males (male-conditioned medium) or wild-type hermaphrodites alone (hermaphrodite-conditioned medium). Both conditioned media contain metabolites secreted by 30,000 young adult worms during 3 hr cultivation (*Figure 1C*). After growing hermaphrodites in the conditioned medium, we found that the hermaphrodites cultured in the male-conditioned medium became paralyzed earlier than those in the hermaphrodite-conditioned medium (80.31% vs. 61.11% paralyzed after 70 min' exposure to aldicarb) (*Figure 1D*). This result suggests that the effect of the male environment on hermaphrodite NMJ is mediated by male-secreted metabolites. In the following experiments, we directly used the male-conditioned medium and hermaphrodite-conditioned medium unless otherwise specified.

We then analyzed muscle excitability as another independent measure of synaptic transmission changes at the NMJ. Previous work has shown that the body-wall muscle at the *C. elegans* NMJ receives both excitatory and inhibitory inputs from cholinergic and GABAergic neurons, respectively (*Richmond and Jorgensen, 1999*). When the excitatory and inhibitory synaptic transmission ratio increases at the NMJ, the excitability of muscle cells should increase. To verify the increased excitability of the body-wall muscle, we expressed the genetically encoded calcium indicator GCaMP3 in muscle cells (under the *myo-3* promoter) and the channelrhodopsin variant Chrimson in VB and DB motor neurons (under the *acr-5* promoter) (*Figure 1E*; *Tian et al., 2009*). Fluorescence changes reflect calcium influx and excitability in the GCaMP3-expressing cells. We found that the baseline GCaMP3 fluorescence is higher in hermaphrodites grown in the male-conditioned medium

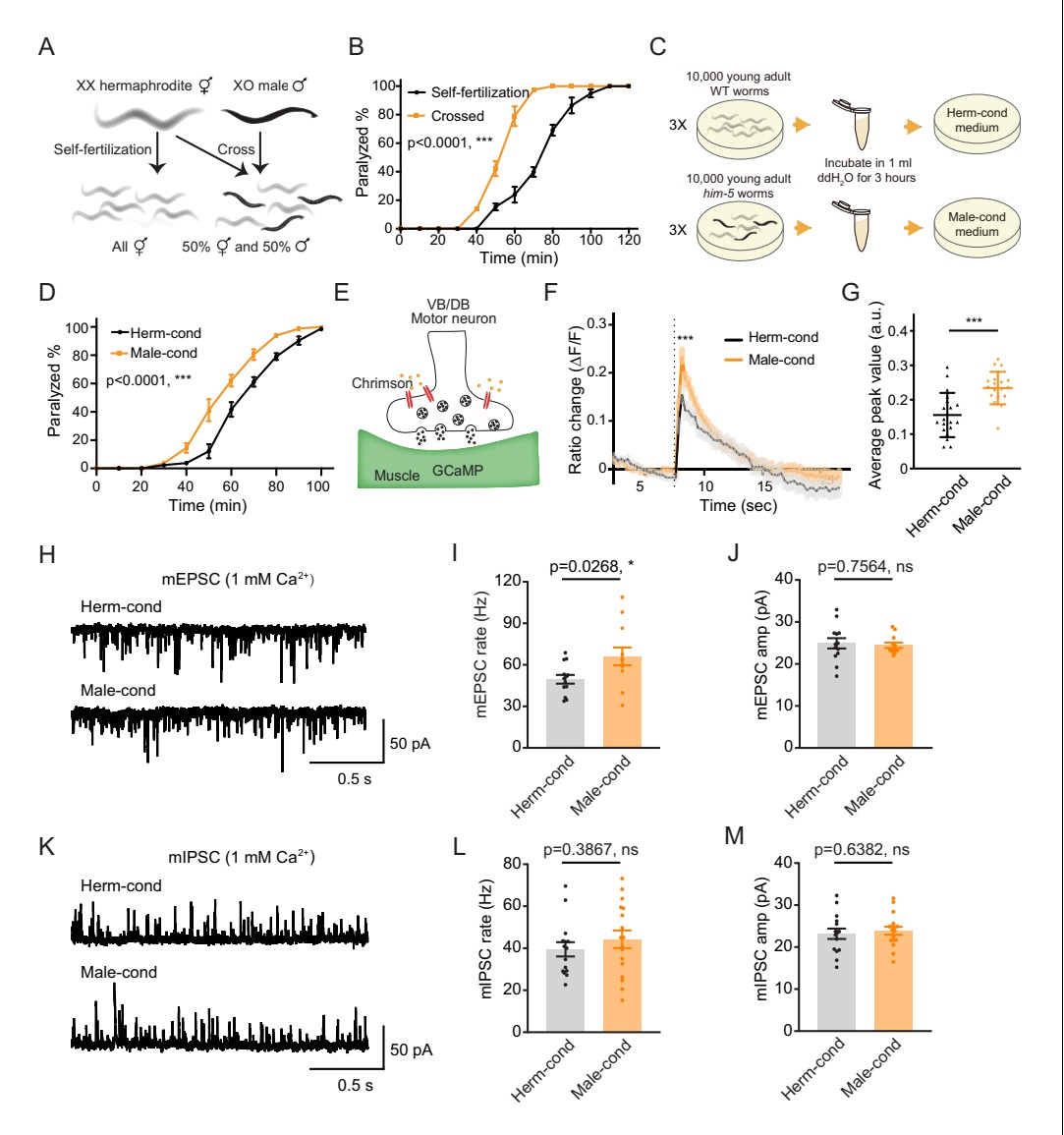

**Figure 1.** The male excretome increases cholinergic synaptic transmission at hermaphrodite NMJ. (A) Schematic illustration of *C. elegans* reproduction. Hermaphrodites with two X chromosomes generate all hermaphrodite progeny via self-fertilization. While hermaphrodites are crossed with males that have a single X chromosome, an equal ratio of hermaphrodite and male offspring are generated. (B) Time course analysis of 1.4 mM aldicarb-induced paralysis in hermaphrodites generated from hermaphrodite self-fertilization (black, Self-fertilization) and hermaphrodite-male crossing (orange, Crossed). (C) Schematic illustration of conditioned medium preparation. 30,000 young-adult wild-type (WT) and *him-5* mutant worms were collected and incubated in 1 ml ddH$_2$O for 3 hr. Metabolites secreted by hermaphrodites and males were collected and used to make hermaphrodite-conditioned (Herm-cond) and male-conditioned (Male-cond) medium. (D) Time course analysis of Aldicarb-induced paralysis in hermaphrodites cultured in hermaphrodite-conditioned medium (black, Herm-cond) and male-conditioned medium (orange, Male-cond). (E) Schematic illustration showing calcium current recording at the *C. elegans* NMJ. Chrimson driven by the *acr-5* promoter was expressed specifically in cholinergic motor neurons, and GCaMP3 under the *myo-3* promoter was expressed in the body-wall muscle. (F–G) Chrimson-evoked calcium transients in body-wall muscle were analyzed using GCaMP3 as a calcium indicator. For adult hermaphrodites cultured in hermaphrodite-conditioned medium (black, Herm-cond) or male-conditioned medium (orange, Male-cond), the averaged responses (F) and the averaged and individual relative increase in GCaMP3 fluorescence intensity ΔF/F (G) are shown. The gray shadings in F indicate the SEM of GCaMP3 responses. The dashed line indicates when the illumination with nominal wavelength at 640 nm for Chrimson activation was applied. (H–J) Endogenous acetylcholine transmission was assessed by recording mEPSCs from body muscles of wild-type adult hermaphrodites cultured in hermaphrodite-conditioned or male-conditioned medium. Representative mEPSC traces (H), the mean mEPSC rates (I), and the mean mEPSC amplitudes (J) are shown. (K–M) Endogenous GABA transmission was assessed by recording mIPSCs from body muscles of wild-type adult hermaphrodites cultured in hermaphrodite-conditioned or male-conditioned medium. Representative mIPSC traces (K), the mean mIPSC rate (L), and the mean mIPSC amplitude (M) are shown. The data for individual animal

*Figure 1 continued on next page*

*Figure 1 continued*

analyzed are indicated. In (**B**), (**D**), (**F–G**), (**I–J**), (**L–M**), *p<0.05, ***p<0.001, ns not significant, two-way ANOVA comparing all of the time points for (**B**) and (**D**), unpaired Student's t-test for (**F–G**), (**I–J**), and (**L–M**).

The online version of this article includes the following source data and figure supplement(s) for figure 1:

**Source data 1.**
**Figure supplement 1.** The physiological muscle excitability is potentiated in hermaphrodites from the male-conditioned medium.
**Figure supplement 1—source data 1.**

compared with those grown in the hermaphrodite-conditioned medium, suggesting relatively higher resting muscle excitability (*Figure 1—figure supplement 1*). Moreover, we excited the VB and DB cholinergic motor neurons via optogenetic activation of Chrimson with red light (wavelength at 640 nm) (*Klapoetke et al., 2014*) and observed significantly increased GCaMP3 fluorescence intensity potentiation (assessed as ΔF/F) in hermaphrodites grown in the male-conditioned medium (*Figure 1F–G*). These results indicate that the male excretome environment causes increased excitatory and inhibitory synaptic transmission ratio and muscle excitability at the NMJ of hermaphrodites.

## The acetylcholine transmission rate at the NMJ is potentiated by the male excretome environment

The increased E/I ratio could be caused by either increased cholinergic transmission or decreased GABAergic transmission. To distinguish between these two possibilities, we analyzed spontaneous miniature excitatory postsynaptic currents (mEPSCs) and miniature inhibitory postsynaptic currents (mIPSCs) at the NMJ. We found that the mEPSC frequency was significantly increased in hermaphrodites from male-conditioned medium compared to those from hermaphrodite-conditioned medium (*Figure 1H,I*), but the mEPSC amplitude was not changed (*Figure 1H,J*). When we examined inhibitory postsynaptic currents, we detected no significant differences in mIPSC frequency and amplitude between hermaphrodites from male- or hermaphrodite-conditioned medium (*Figure 1K–M*). The electrophysiology data suggest that potentiation of acetylcholine transmission rate mainly contributes to the observed increase in the E/I ratio at the NMJ of hermaphrodites in the male excretome environment.

## The male excretome environment increases the hermaphrodite NMJ synaptic transmission during the juvenile stage

To delineate whether there are any critical developmental windows for the observed synaptic transmission modulation by the male environment, we transferred hermaphrodites to male-conditioned medium at a series of different developmental stages (egg, L1 [24 hr after egg], L2–L3 [36 hr after egg], and mid-L4 [48 hr after egg]). We then measured synaptic transmission in young adults with the aldicarb assay (*Figure 2A*). The hermaphrodites transferred to the male-conditioned medium at the egg, L1, and L2–L3 stage presented significantly increased sensitivity to aldicarb when they grow into young adult (*Figure 2B*, *Figure 2—figure supplement 1A–C*, 92.5% vs. 55.1% at 70 min for egg stage, 46.9% vs. 24.9% for the L1 stage, and 71.3% vs. 14.0% for L2–L3 stage). In contrast, we observed no differences in sensitivity to aldicarb between hermaphrodites transferred to male-conditioned medium at the mid-L4 stage and those from hermaphrodite-conditioned medium (*Figure 2B*, *Figure 2—figure supplement 1D*, 15.9% vs. 14.0% at 70 min). Those data suggest that exposure to the male excretome environment in L3–L4 stage is critical for modulation of the NMJ synaptic transmission in hermaphrodites.

To study whether the sustained male environment is required to maintain the cholinergic synaptic transmission potentiation at NMJ, we removed hermaphrodites from male-conditioned medium out of the male environment at L4 (48 hr after egg) and young adult (60 hr after egg). 24 and 12 hr later, we performed the aldicarb assay (*Figure 2A,C*, *Figure 2—figure supplement 1E*). We found that hermaphrodites leaving the male excretome environment at the young adult stage still showed an increased sensitivity to aldicarb compared with those in the hermaphrodite-conditioned medium (66.3% vs. 42.0% at 70 min). The effect was comparable to that in hermaphrodites sustained in the male environment (78.0% vs. 42.0%) (*Figure 2C*), suggesting that the maintenance of the elevated

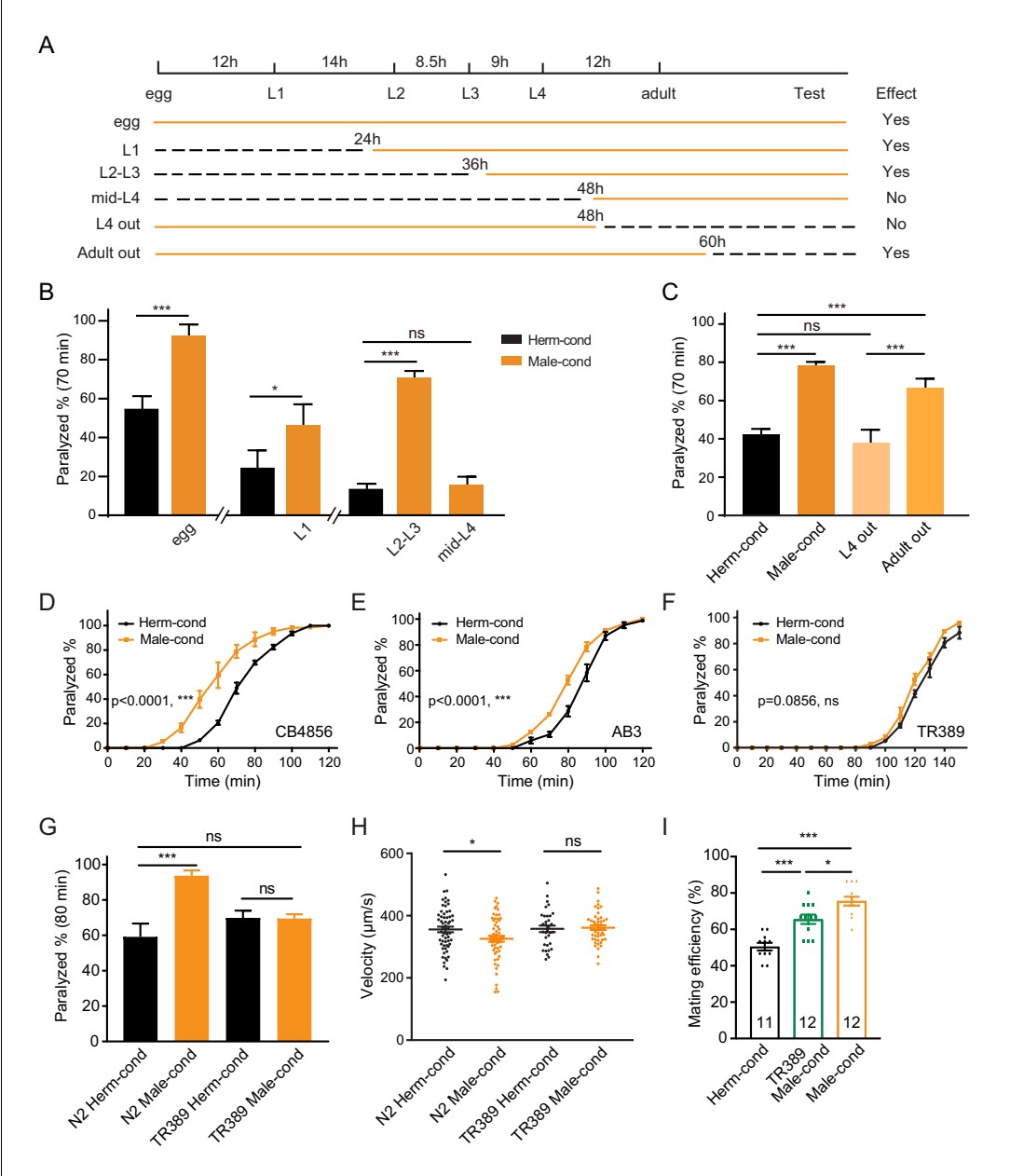

**Figure 2.** The male excretome modulates the hermaphrodite NMJ synaptic transmission in a developmental-stage-dependent manner. (**A**) Schematic illustration of the life cycles of *C. elegans* and the time when the hermaphrodite-conditioned medium (dashed black lines) or male-conditioned medium (solid orange lines) was applied. (**B**) The percentage of animals paralyzed on 1.4 mM aldicarb at 70 min were plotted for hermaphrodites cultured in male-conditioned medium (orange) starting from the egg stage, L1 stage, L2–L3 stage, and mid-L4 stage. Hermaphrodites cultured in hermaphrodite-conditioned medium (black) served as controls. (**C**) The percentage of animals paralyzed on 1.4 mM aldicarb at 70 min were plotted for hermaphrodites cultured in male-conditioned medium from the egg stage to the mid-L4 stage (L4 out) and young adult stage (Adult out); hermaphrodites cultured in hermaphrodite-conditioned medium (black) or male-conditioned medium (orange) served as controls. (**D–F**) Time course analysis of aldicarb-induced paralysis in hermaphrodites cultured in hermaphrodite-conditioned medium (black) and male-conditioned medium (orange) in CB4856 (**D**), AB3 (**E**), and TR389 (**F**) strains. (**G**) The percentage of animals paralyzed on 1.4 mM aldicarb at 80 min were plotted for N2 hermaphrodites cultured in N2 hermaphrodite (N2 herm-cond)-, N2 male (N2 Male-cond)-, TR389 hermaphrodite (TR389 Herm-cond)- or TR389 male (TR389 Male-cond)-conditioned medium. (**H**) Locomotion behavior analysis of single adult hermaphrodite cultured in N2 hermaphrodite (Herm-cond)-, TR389 male (TR389 Male-cond)-, and N2 male (Male-cond)-conditioned medium. The averaged and individual crawling locomotion velocities were plotted. (**I**) Measurement of hermaphrodite mating efficiency cultured in N2 hermaphrodite-, TR389 male-, and N2 male-conditioned medium. In B-I, *p<0.05, ***p<0.001, ns not significant, two-way ANOVA with post hoc Sidak multiple comparisons for (**B–C**) and (**G**), two-way ANOVA comparing all of the time points for (**D–F**), one-way ANOVA with post hoc Dunnett multiple comparisons for ( **H** and **I**).

*Figure 2 continued on next page*

*Figure 2 continued*

The online version of this article includes the following source data and figure supplement(s) for figure 2:

**Source data 1.**
**Figure supplement 1.** The male environment modulates the hermaphrodite NMJ synaptic transmission in a developmental-stage-dependent manner.
**Figure supplement 1—source data 1.**
**Figure supplement 2.** TR389 hermaphrodites can be modulated by the modulator ascarosides.
**Figure supplement 2—source data 1.**
**Figure supplement 3.** The male excretome does not change hermaphrodite body-bend curvature.
**Figure supplement 3—source data 1.**
**Figure supplement 4.** The male excretome does not modulate hermaphrodite brood size.
**Figure supplement 4—source data 1.**

cholinergic synaptic transmission rate at the hermaphrodites NMJ does not require a sustained male excretome environment in adults. In contrast, we observed that the hermaphrodite leaving the male-conditioned medium at the mid-L4 stage presented similar aldicarb sensitivity to those from the hermaphrodite-conditioned medium (37.7% vs. 42.0% at 70 min) (*Figure 2C*, *Figure 2—figure supplement 1E*). Taken together, these data support the notion that the male environment exposure at a critical period (the L3–L4 stage) is required for the modulation of hermaphrodites NMJ cholinergic synaptic transmission.

The aforementioned experiments were carried out using the Bristol N2 strain. To determine whether the male excretome environment's effect is conserved in other *C. elegans* strains, we studied several natural variations, including the Australian strain AB3, the Hawaiian strain CB4856, and the Madison strain TR389. We observed that the male-conditioned medium accelerated animal paralysis in the CB4856 (*Figure 2D*, 59.7% vs. 20.7% after 60 min of aldicarb) and the AB3 strains (*Figure 2E*, 52.8% vs. 28.6% after 80 min of aldicarb), but not in the TR389 strain (*Figure 2F*, 68.46% vs. 60.99% after 130 min of aldicarb). Thus, although the effect of the male environment does exist in other natural *C. elegans* strains, exceptions do exist, as in the TR389 strain.

Two possibilities could account for this lack of a modulator effect: TR389 males may not be able to secrete the modulator ascarosides; alternatively, TR389 hermaphrodites cannot sense and respond to the modulator ascarosides. To distinguish between these two possibilities, we grew Bristol N2 hermaphrodites in the TR389 male-conditioned medium and compared their synaptic transmission by aldicarb sensitivity with those maintained in N2 and TR389 hermaphrodite-conditioned medium. The three groups presented similar sensitivity to aldicarb (*Figure 2G*). In contrast, TR389 hermaphrodites grown in the N2 male-conditioned medium showed significantly increased sensitivity to aldicarb compared to those in the N2 hermaphrodite-conditioned medium (*Figure 2—figure supplement 2*). Thus, TR389 males appear unable to secrete the modulator ascarosides.

## The male excretome environment alters hermaphrodite locomotion and promotes mating efficiency

As mentioned above, the coordination of excitatory and inhibitory innervations at NMJ guarantees *C. elegans* sinusoidal movement. To study whether the altered cholinergic synaptic transmission impacts body-bend amplitude and coordination of animal movement, we compared the locomotion of hermaphrodites from male- or hermaphrodite-conditioned medium. We observed that males had higher body curvature and locomotor velocity than hermaphrodites (*Figure 2—figure supplement 3A–B*), consistent with previous studies (*Mowrey et al., 2014*). We did not observe body-bend curvature differences in hermaphrodites from male- and hermaphrodite-conditioned medium (*Figure 2—figure supplement 3C–D*). However, the locomotor velocities of hermaphrodites from male-conditioned medium are significantly lower than those from hermaphrodite-conditioned medium (*Figure 2H*). In contrast, the TR389 male-conditioned medium did not show similar effects (*Figure 2H*). This supports the notion that the altered NMJ synaptic transmission by the male excretome affects hermaphrodite locomotion. It is possible that the disturbance of excitatory and inhibitory synaptic transmission balance at NMJ compromise locomotion activity.

Communications between conspecifics modulate behaviors and alter physiology to allow appropriate responses to particular social environments. To study the physiological significance of male

excretome modulation, we tested its effect on hermaphrodite's egg-laying behaviors and mating abilities. We calculated the brood size of hermaphrodites from hermaphrodite- and male-conditioned medium, and observed no significant differences between the two groups (*Figure 2—figure supplement 4*). Then we measured the mating efficiency with males in hermaphrodites from male- and hermaphrodite-conditioned medium. Two young adult stage hermaphrodites from male- or hermaphrodite-conditioned medium were cultured with two young-adult males for 24 hr. Successful mating was scored when more than three male progenies were generated in the mating plate. The results showed that hermaphrodites from TR389 male-conditioned medium had higher mating efficiency compared to those from hermaphrodite-conditioned medium (*Figure 2I*), which is consistent with previous research that the male environment reduces hermaphrodite exploration and increases mating behaviors (*Aprison and Ruvinsky, 2019a*; *Aprison and Ruvinsky, 2019b*). Interestingly, we found that N2 male-conditioned medium showed a significant further increase of hermaphrodite mating efficiency than the TR389 male-conditioned medium (*Figure 2I*). We speculate that the N2 males secrete additional metabolites to modulate locomotion and mating efficiency in hermaphrodites.

## Male-specific ascarosides mediate the modulatory effect of the male excretome environment on the hermaphrodite NMJ synaptic transmission

To identify the additional metabolites secreted by the N2 males, we focused on searching the male pheromones. In *C. elegans*, ascarosides are known to function as pheromones to mediate social interactions and modulate development (*Butcher et al., 2007*; *Butcher et al., 2009*; *Ludewig et al., 2019*; *Ludewig and Schroeder, 2013*; *Srinivasan et al., 2008*; *Wu et al., 2019*). We hypothesized that the observed effects of the male environment on hermaphrodite cholinergic synaptic transmission at the NMJ may be mediated by male-specific ascarosides. Ascarosides are derivatives of 3,6-dideoxysugar ascarylose, and their biosynthesis requires several dehydrogenases, including DAF-22, which β-oxidizes and shortens long-chain fatty acids to generate bioactive medium- and short-chain ascarosides (*Figure 3A*; *Butcher et al., 2009*; *von Reuss et al., 2012*; *Zhou et al., 2018*). Therefore, most of the active short- and medium-chain ascarosides are absent from the metabolomes of *daf-22* mutants (*Butcher et al., 2009*; *von Reuss et al., 2012*; *Zhou et al., 2018*).

To test whether the effect of the male environment on hermaphrodite NMJ synaptic transmission is mediated by ascarosides, we grew hermaphrodites in *daf-22* conditioned medium and compared their aldicarb sensitivity to hermaphrodites grown in the wild-type conditioned medium. We found that the hermaphrodites grown in the *daf-22* male-conditioned medium exhibited similar aldicarb sensitivity with those grown in *daf-22* hermaphrodite-conditioned medium (*Figure 3B*, 43.7% vs. 43.7% at 70 min). The inability of the *daf-22* male environment to modulate hermaphrodite NMJ synaptic transmission suggests that male-specific ascarosides do contribute to the observed modulatory effects on synaptic transmission.

The pheromone effects are often sexually dimorphic. To study whether the male-specific ascarosides also modulate male NMJ synaptic transmission, we compared the aldicarb sensitivity of males grown in the hermaphrodite- and male-conditioned media. We did not observe aldicarb sensitivity differences in males from hermaphrodite- and male-conditioned medium (*Figure 3—figure supplement 1A*). Since males can secrete those modulatory pheromones themselves, we took advantage of *daf-22* mutant males that have the defects in pheromone production. We also found that *daf-22* males from the male-conditioned medium did not show any significant differences in aldicarb sensitivity compared to those from the hermaphrodite-conditioned medium (*Figure 3C*). In contrast, the *daf-22* hermaphrodites showed higher aldicarb sensitivity from the male-conditioned medium compared to those from the hermaphrodite-conditioned medium (*Figure 3—figure supplement 1B*), suggesting *daf-22* mutation did not alter the modulatory effect of male-conditioned medium on hermaphrodites. These results indicate that male-specific ascarosides cannot modulate synaptic transmission in males, suggesting a sexually dimorphic effect of those male-specific ascarosides.

The *C. elegans* ascarosides comprise a complex mixture of ascaroside derivates that vary according to their side chains; there are saturated, α,β-unsaturated (e.g., α,β double-bond), and β-hydroxylated (e.g., β-hydroxylated side chain) derivatives. Hermaphrodites and males are known to accumulate distinct types and quantities of these various ascarosides (*Butcher et al., 2009*; *von Reuss et al., 2012*). To identify the 'modulator ascarosides' that function in the observed

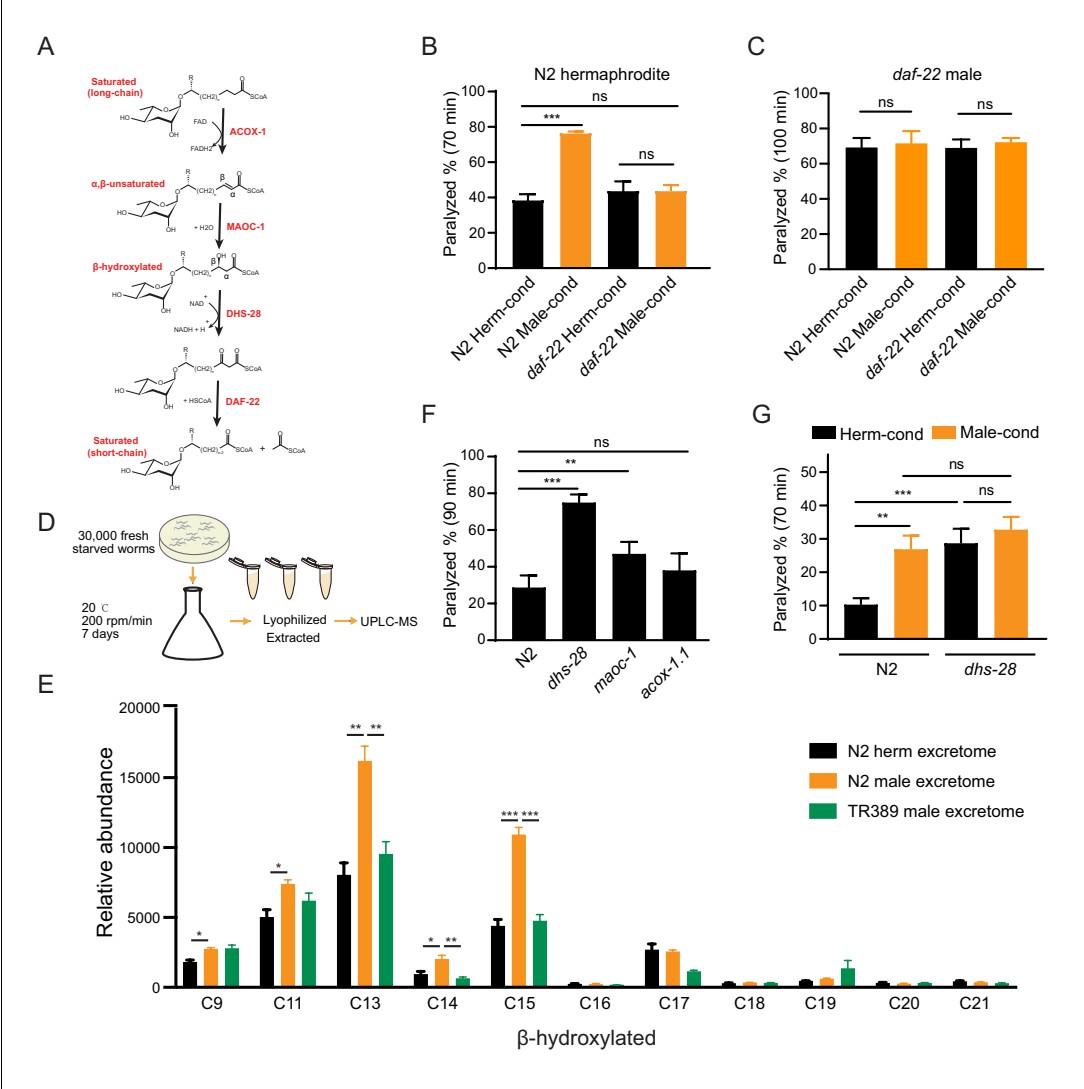

**Figure 3.** Ascarosides in the male environment modulate hermaphrodite NMJ synaptic transmission. (**A**) Proposed model of peroxisomal β-oxidation enzymes ACOX-1, MAOC-1, DHS-28, and DAF-22 in ascaroside side-chain biosynthesis. (**B**) The percentage of animals paralyzed on 1.4 mM aldicarb at 70 min were plotted for N2 hermaphrodites cultured in hermaphrodite (N2 Herm-cond)-, male (N2 Male-cond)-, *daf-22* mutants herm (*daf-22* Herm-cond)-, or *daf-22* mutant male (*daf-22* Male-cond)-conditioned medium. (**C**) The percentage of animals paralyzed on 0.5 mM aldicarb at 100 min were plotted for *daf-22* mutant males cultured in hermaphrodite (N2 Herm-cond)-, male (N2 Male-cond)-, *daf-22* mutants herm (*daf-22* Herm-cond)-, or *daf-22* mutant male (*daf-22* Male-cond)-conditioned medium. (**D**) Schematic illustration of excretome preparation for UPLC-MS. Around 30,000 freshly starved worms were cultured in S medium supplemented with concentrated OP50 for 7 days. The excretomes were collected by centrifugation, filtration, and lyophilized extraction, followed by UPLC-MS analysis. (**E**) β-hydroxylated ascaroside profiles in excretomes obtained from N2 hermaphrodites (N2 herm excretome), N2 mixed-gender animals of *him-5* mutants (N2 male excretome), and TR389 mixed-gender animals (TR389 male excretome). (**F**) The percentage of animals paralyzed on 1.4 mM aldicarb at 90 min were plotted for β-oxidation mutants (*acox-1.1*, *maoc-1*, and *dhs-28*). (**G**) The percentage of animals paralyzed on 1.4 mM aldicarb at 70 min were plotted for N2 and *dhs-28* mutant hermaphrodites cultured in hermaphrodite-conditioned medium (Herm-cond), male-conditioned medium (Male-cond). In (**B–C**), (**E–G**), *p<0.05, **p<0.01, ***p<0.001, ns not significant, two-way ANOVA with post hoc Sidak multiple comparisons for (**B–C**) and (**F–G**). one-way ANOVA with post hoc Dunnett multiple comparisons for (**E**).

The online version of this article includes the following source data and figure supplement(s) for figure 3:

**Source data 1.**

**Figure supplement 1.** The male environment cannot modulate NMJ synaptic transmission in males.

**Figure supplement 1—source data 1.**

**Figure supplement 2.** UPLC-MS analysis of excretome from animal cultures.

**Figure supplement 2—source data 1.**

modulation of the hermaphrodite NMJ synaptic transmission, we first analyzed the TR389 strain, which recalls appearing unable to secrete the modulator ascarosides.

To further determine the identity of the modulator ascarosides, we used ultra-performance liquid chromatography–mass spectrometry (UPLC-MS) analyses to compare the excretomes among N2 hermaphrodite cultures (containing N2 hermaphrodites only), N2 male cultures, *daf-22* male cultures, and TR389 male cultures (all of the male cultures contains around 35% males) (*Figure 3D*). We collected and analyzed culture media samples with UPLC-MS and found that ascr#10 was enriched in both the N2 male and TR389 male cultures relative to the N2 hermaphrodite cultures (*Figure 3—figure supplement 2A*), consistent with previous reports (*Izrayelit et al., 2012*). We also observed that the *daf-22* male cultures lacked most of the short- and medium-chain ascarosides, and accumulated the long-chain ascarosides (*Figure 3—figure supplement 2B*), confirming the role of DAF-22 in dehydrogenating and shortening ascaroside side chains.

Next, reasoning that the modulator ascarosides should be enriched in N2 male-conditioned culture, we compared the UPLC-MS profiles of N2 male cultures with the N2 hermaphrodites and the TR389 male cultures. The medium-chain β-hydroxylated ascarosides were substantially increased in the N2 male cultures compared to the N2 hermaphrodite cultures and TR389 male cultures. Specifically, the significantly enriched β-hydroxylated ascarosides in N2 males included C13, C14, and C15 ascarosides (*Figure 3E*). Notably, we detected no significant changes between the N2 and TR389 male cultures for saturated ascarosides (*Figure 3—figure supplement 2C*). These results implicate that the medium-chain β-hydroxylated ascarosides may act as the male modulator ascarosides.

Pursuing this with a genetic approach, we acquired a mutant of the known ascaroside synthesis enzyme DHS-28; previous analysis of the *dhs-28* mutant hermaphrodite metabolome has shown that these animals accumulate β-hydroxylated medium-chain ascarosides (*Butcher et al., 2009*; *von Reuss et al., 2012*). We conducted aldicarb assays to compare the E/I ratios of *dhs-28* cultures with those of N2 hermaphrodites grown in the hermaphrodite-conditioned medium. As expected, *dhs-28* mutant hermaphrodites were more sensitive to aldicarb compared with N2 hermaphrodites (*Figure 3F*). We also tested hermaphrodites of other known ascaroside synthesis mutants, including *maoc-1* and *acox-1.1* – which are known to accumulate saturated and α,β-unsaturated side-chain ascarosides – but found that *maoc-1* mutants present slightly increase of aldicarb sensitivity than the wild type, and *acox-1.1* mutants were indifferent from the wild type for their sensitivity to aldicarb (*Figure 3F*). Furthermore, we examined the male ascaroside effects on *dhs-28* mutants, which could accumulate β-hydroxylated medium-chain ascarosides themselves. The result showed that *dhs-28* hermaphrodites cannot be modulated by male ascarosides by presenting comparable aldicarb sensitivity when in hermaphrodite- and male-conditioned medium (*Figure 3G*). These experiments with ascaroside biosynthesis mutants establish that environmental enrichment of β-hydroxylated medium-chain ascarosides increases the hermaphrodite NMJ E/I ratio, thereby supporting that these specific ascarosides may function as NMJ cholinergic synaptic transmission modulators.

## AWB sensory neurons are involved in sensing the modulator ascarosides and transmit signals to the NMJ through cGMP signaling

Pheromone signals in the environment are detected and integrated by chemosensory neural circuits (*Ludewig and Schroeder, 2013*; *Srinivasan et al., 2008*). In *C. elegans*, there are 11 pairs of chemosensory neurons that can respond to pheromone signals (ASE, AWC, AWA, AWB, ASH, ASI, ADF, ASG, ASJ, ASK, and ADL). To identify the specific chemosensory neurons sensing the modulator ascarosides, we used a miniSOG (mini Singlet Oxygen Generator)-induced genetic ablation strategy. miniSOG is an engineered fluorescent protein that can generate singlet oxygen upon blue light illumination. Targeting miniSOG to mitochondria can lead to singlet oxygen accumulation in mitochondria, which induces rapid and efficient cell death (*Qi et al., 2012*). To examine the genetic ablation efficiency, we coexpressed mCherry and miniSOG in the chemosensory neurons under the control of the *flp-21* promoter and quantified the miniSOG ablation efficiency based on the percentage of live neurons labeled by mCherry before and after induction of cell death. To optimize the ablation protocol, we tested continuous blue light stimulation at a power of 57 mW/cm$^2$ over different periods (*Figure 4—figure supplement 1A*). We found that 15 min' stimulation resulted in complete loss of mCherry signals in around 47.8% of neurons and a dramatic reduction of mCherry signals in 26.1% of neurons, whereas stimulation for 30 min or 50 min led to complete loss of mCherry signals in almost 80% of neurons and faint residual expression of mCherry signals in 20% of neurons

(*Figure 4—figure supplement 1B,C*). Considering both the ablation efficiency and the stimulation time, we chose 30 min of continuous blue light stimulation for our standard ablation procedure.

We screened all the 11 pairs of chemosensory neurons based on miniSOG-induced genetic ablation of hermaphrodites at the late L1 stage (*Figure 4A*). We grew hermaphrodites in male-conditioned or hermaphrodite-conditioned medium following ablation of each specific chemosensory neuron type, and measured their sensitivity to aldicarb. Ablation of the AWB (*str-1* promoter driving miniSOG) neuron pair in hermaphrodites blocked the increased sensitivity to aldicarb following exposure to the male-conditioned medium (*Figure 4B,C*, 45.6% vs. 43.4% at 80 min). In contrast, the increased aldicarb sensitivity in the male-conditioned medium remained when any other chemosensory neurons were ablated (*Figure 4A*, *Figure 4—figure supplement 2A–G*). To further confirm the requirement of AWB in sensing the modulator ascarosides, we compared the locomotion of AWB-ablated hermaphrodites from male- and hermaphrodite-conditioned medium. Our data showed that ablation of AWB neurons decreased the locomotion velocity in hermaphrodites from the hermaphrodite-conditioned medium. In addition, ablation of AWB neurons blocked the decreased velocity by the modulator ascarosides (*Figure 4D*). These results support that AWB neurons in hermaphrodites are necessary for the effects of male-specific modulator ascarosides on NMJ synaptic transmission.

We also tested whether activation of AWB neurons is sufficient to modulate NMJ synaptic transmission. We specifically expressed the channelrhodopsin variant CHIEF in AWB neurons and administered blue light illumination in the presence of all-trans retinal (ATR) to activate AWB neurons throughout the L4 stage. We observed decreased aldicarb sensitivity in animals fed with ATR (ATR+ light- vs. ATR− light−). Nevertheless, hermaphrodites with activated AWB neurons during the L4 stage showed higher sensitivity to aldicarb than controls without blue light activation (ATR+ light+ vs. ATR+ light−) (*Figure 4E*). This effect is absent in the groups lacking ATR (ATR− light+ vs. ATR− light−) (*Figure 4E*). In contrast, activation of ASJ/ASI neurons or other amphid wing neurons like AWA and AWC cannot increase aldicarb sensitivity, and the hermaphrodites with AWA neurons activation even present slightly decreased aldicarb sensitivity (*Figure 4—figure supplement 3*). These findings confirm that activation of the AWB chemosensory neuron pair in hermaphrodites is sufficient to modulate the NMJ synaptic transmission.

In order to test whether AWB neurons directly sense those modulator ascarosides, we monitored intracellular $Ca^{2+}$ dynamics upon male excretomes stimulation by expressing the calcium indicator GCaMP6f in AWB neurons. We found that the AWB neurons elicited a rapid and robust calcium transient responding to the male excretomes. However, no responses were detected by stimulation with the hermaphrodite excretomes (*Figure 4F–H*). Collectively, the data support that AWB neurons directly respond to the male-specific modulator ascarosides.

We next explored which signaling molecules in AWB neurons mediate their responsivity to the modulator ascarosides. In *C. elegans*, most chemical odors are perceived upon their binding to specific G-protein coupled receptors (GPCRs) located in chemosensory neurons; these receptors subsequently activate downstream signaling cascades (*Bargmann, 2006*; *Li and Liberles, 2015*; *Liberles, 2014*). The G protein ODR-3 and the cGMP-gated channels TAX-2 have been implicated in chemosensory signal transduction in AWB neurons. We first examined the NMJ E/I ratio in *tax-2* and *odr-3* mutant hermaphrodites. We observed no differences in sensitivity to aldicarb for *tax-2* mutant hermaphrodites upon exposure to the male-conditioned or hermaphrodite-conditioned media (*Figure 4I*). In the *odr-3* mutants, we even observed a decreased aldicarb sensitivity in hermaphrodites from male-conditioned medium (*Figure 4J*), suggesting that the ability to mediate the downstream signaling effects of modulator ascarosides and increase NMJ E/I ratio is disrupted in these mutants (*Figure 4I,J*). Further supporting this, complementing TAX-2 expression in AWB neurons rescued the increased aldicarb sensitivity phenotype of hermaphrodites grown in the male-conditioned medium (*Figure 4I*). Expression of TAX-2 in ASI and ASJ neurons had no such rescue effect (*Figure 4I*). The increased sensitivity to aldicarb was also rescued by ODR-3 complementation in AWB neurons (*Figure 4J*). Together, these results establish that the cGMP signaling pathway in AWB chemosensory neurons transmits male-specific modulator ascaroside signals to the NMJ.

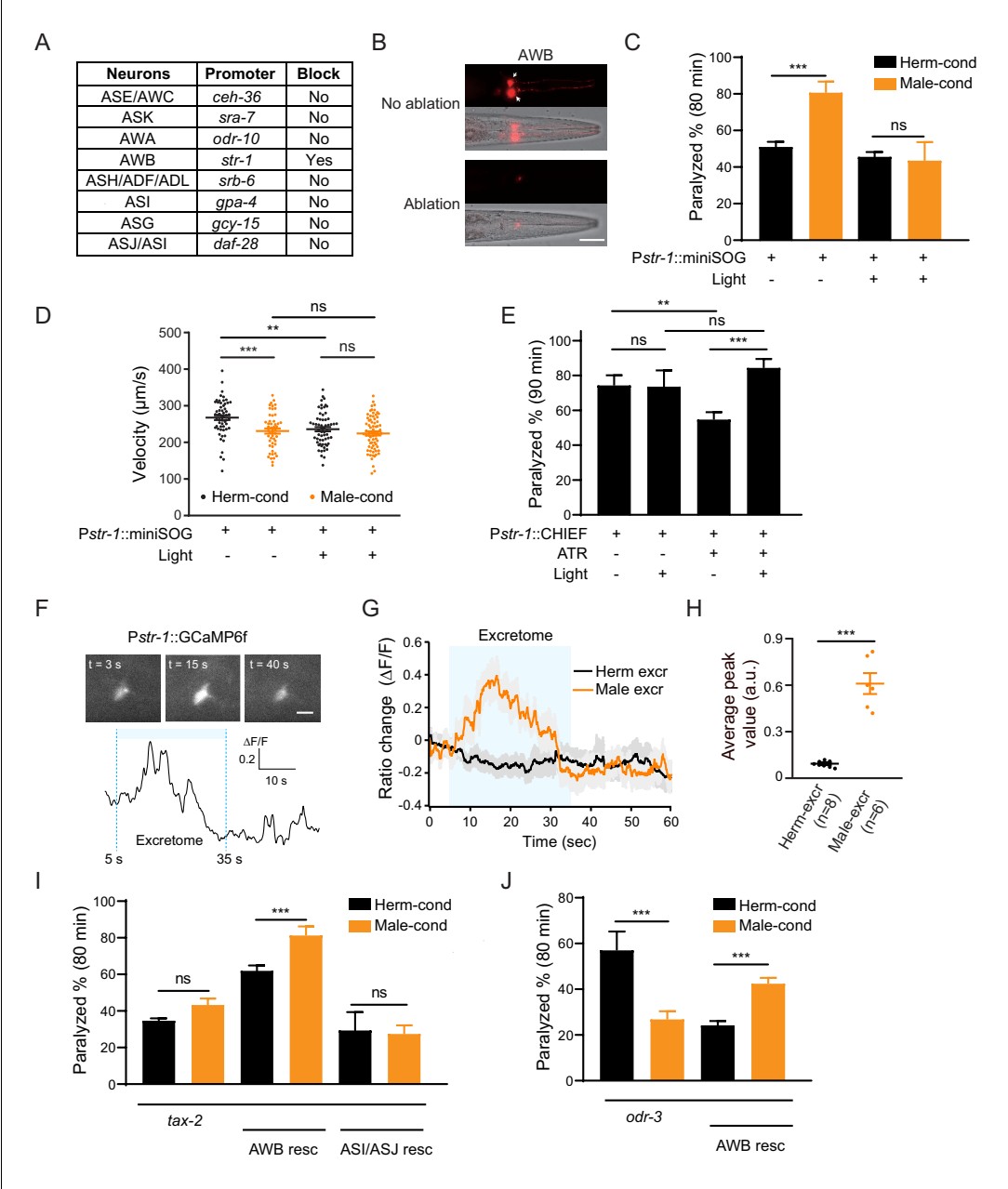

**Figure 4.** AWB neurons sense the modulator ascarosides. (**A**) The table lists all of the chemosensory neurons examined in the screen, the neuron-specific promoters used to drive miniSOG expression, and the impact of neuron ablation on sensing of modulator ascarosides. (**B**) Representative images showing that mCherry-labeled AWB neurons were specifically ablated by blue light-induced miniSOG activation. Scale bar, 40 μm. (**C**) The percentage of animals paralyzed on 1.4 mM aldicarb at 80 min were plotted for hermaphrodites expressing miniSOG in AWB neurons (*str-1* promoter) with and without blue-light-induced ablation. Black: cultured in hermaphrodite-conditioned medium; Orange: cultured in male-conditioned medium. (**D**) Locomotion behavior analysis of single adult worm from AWB-ablated hermaphrodites in hermaphrodite- and male-conditioned medium. The averaged crawling locomotion velocities were plotted. (**E**) The percentage of animals paralyzed on 1.4 mM aldicarb at 90 min were plotted for hermaphrodites with AWB neurons optogenetically activated during the L4 stage. The channelrhodopsin variant CHIEF was expressed in AWB chemosensory neurons driven by the *str-1* promoter. The blue light was turned on to excite AWBs in transgenetic animals fed with or without ATR. (**F**) Top: snapshots of GCaMP6f fluorescent signals of an AWB neuron before, during, and after addition of male excretome. Scale bar, 10 μm. Bottom: the calcium trace showing the AWB neuron activated by male excretome. (**G**) Curves and average intensities of $Ca^{2+}$ signals evoked by the hermaphrodite or male excretome in the soma of AWB with GCaMP6f expression. The shaded box represents the addition of the hermaphrodite or male excretome. (**H**) Scatter diagram and quantification of the $Ca^{2+}$ change. Each point represents $Ca^{2+}$ peak value from one animal. (**I**) The percentages of animals paralyzed on 1.4 mM aldicarb at 80 min were plotted for *tax-2(p691)* mutant hermaphrodites and TAX-2 expression restored in AWB or ASJ/ASI neurons cultured in hermaphrodite- (black) and male-conditioned medium (orange). (**J**) The percentages of animals paralyzed on 1.4 mM aldicarb at 80

*Figure 4 continued on next page*

*Figure 4 continued*

min were plotted for *odr-3(n1605)* mutant hermaphrodites and ODR-3 expression restored in AWB neurons cultured in hermaphrodite- (black) and male-conditioned medium (orange). In (C–E), (H–J), ***p<0.001, **p<0.01, ns not significant, two-way ANOVA with post hoc Sidak multiple comparisons for (C), (E), and (I–J), one-way ANOVA with post hoc Dunnett multiple comparisons for (D), unpaired Student's t-test for (H).

The online version of this article includes the following source data and figure supplement(s) for figure 4:

**Source data 1.**
**Figure supplement 1.** Genetic ablation efficiency by miniSOG.
**Figure supplement 1—source data 1.**
**Figure supplement 2.** ASE, AWC, AWA, ASH, ASI, ADF, ASG, ASK, ADL, and ASJ chemosensory neurons are dispensable for sensing modulator ascarosides.
**Figure supplement 2—source data 1.**
**Figure supplement 3.** Activation of other sensory neurons does not change aldicarb sensitivity in hermaphrodites.
**Figure supplement 3—source data 1.**

## Excitatory postsynaptic receptor clustering is increased in hermaphrodites exposed to the male environment

The steps of the synaptic transmission process include presynaptic vesicle fusion, neurotransmission, and neurotransmitter binding to postsynaptic receptors. The increased cholinergic synaptic transmission rate at the hermaphrodite NMJ induced by the modulator ascarosides could reflect changes in any of these steps. We first examined whether any NMJ synaptic structures were altered, specifically by labeling cholinergic synapses via expression of a RAB-3-GFP fusion protein in DA and DB cholinergic motor neurons (using the *unc-129* promoter) (*Colavita et al., 1998*). DA and DB neurons are known to receive synaptic inputs in the ventral nerve cord and to form NMJs with the body-wall muscle in the dorsal nerve cord, and this results in the formation of puncta comprising presynaptic RAB-3 proteins that can be observed at DA/DB axon terminals in the dorsal cord (*Colavita et al., 1998*). We observed that puncta fluorescence intensities and densities were comparable in hermaphrodites grown in either hermaphrodite- or male-conditioned medium (*Figure 5A*), which suggested that the excitatory synapse structures were unaltered by the presence of modulator ascarosides. We next labeled the GABAergic motor neuron terminals by expressing RAB-3 fused with RFP under the *unc-25* promoter (*Jin et al., 1999*). Similar to the excitatory cholinergic synapses, the puncta fluorescence intensities and densities in the inhibitory GABAergic synapses did not differ between hermaphrodites from male-conditioned or hermaphrodite-conditioned medium (*Figure 5B*), which collectively suggest that neither excitatory nor inhibitory synapse structures are affected by the modulator ascarosides.

We then examined the extent of excitatory and inhibitory postsynaptic receptor localization in hermaphrodites by analyzing the subcellular distributions of nicotinic acetylcholine receptors (nAchRs; excitatory) and GABA$_A$ receptors (inhibitory). A single-copy transgenic insertion technique was applied to express fluorescence reporter fusion variants of two known nAchR subunit proteins (UNC-29-RFP and ACR-16-RFP) or a GABA$_A$ receptor subunit (UNC-49-mCherry) under the control of a muscle-specific promoter. At cholinergic synapses, the hermaphrodites from the male-conditioned medium had a slight but significant increase in puncta signal intensities for the nAchRs compared to those from the hermaphrodite-conditioned medium (*Figure 5C,D*). However, the GABA$_A$R intensities were not changed (*Figure 5E*). Both the nAchRs and GABA$_A$R densities were unaltered (*Figure 5C–E*). Thus, the male-specific modulator ascarosides are involved in increased postsynaptic receptor abundance at excitatory synapses in hermaphrodites.

## Presynaptic CaV2 calcium channel localization at NMJ cholinergic synapses is increased in hermaphrodites exposed to the male environment

Next, we examined if the process of presynaptic neurotransmission is regulated based on the fact that the mEPSC frequency was increased in hermaphrodites by the male-specific pheromones (*Figure 1H–J*). N-type voltage-gated calcium channels (CaV2) are required for the presynaptic calcium influx process that underlies both excitatory and inhibitory neurotransmission (*Liu et al., 2018*; *Tong et al., 2017*). Therefore, we inspected CaV2 calcium channel localization and abundance at

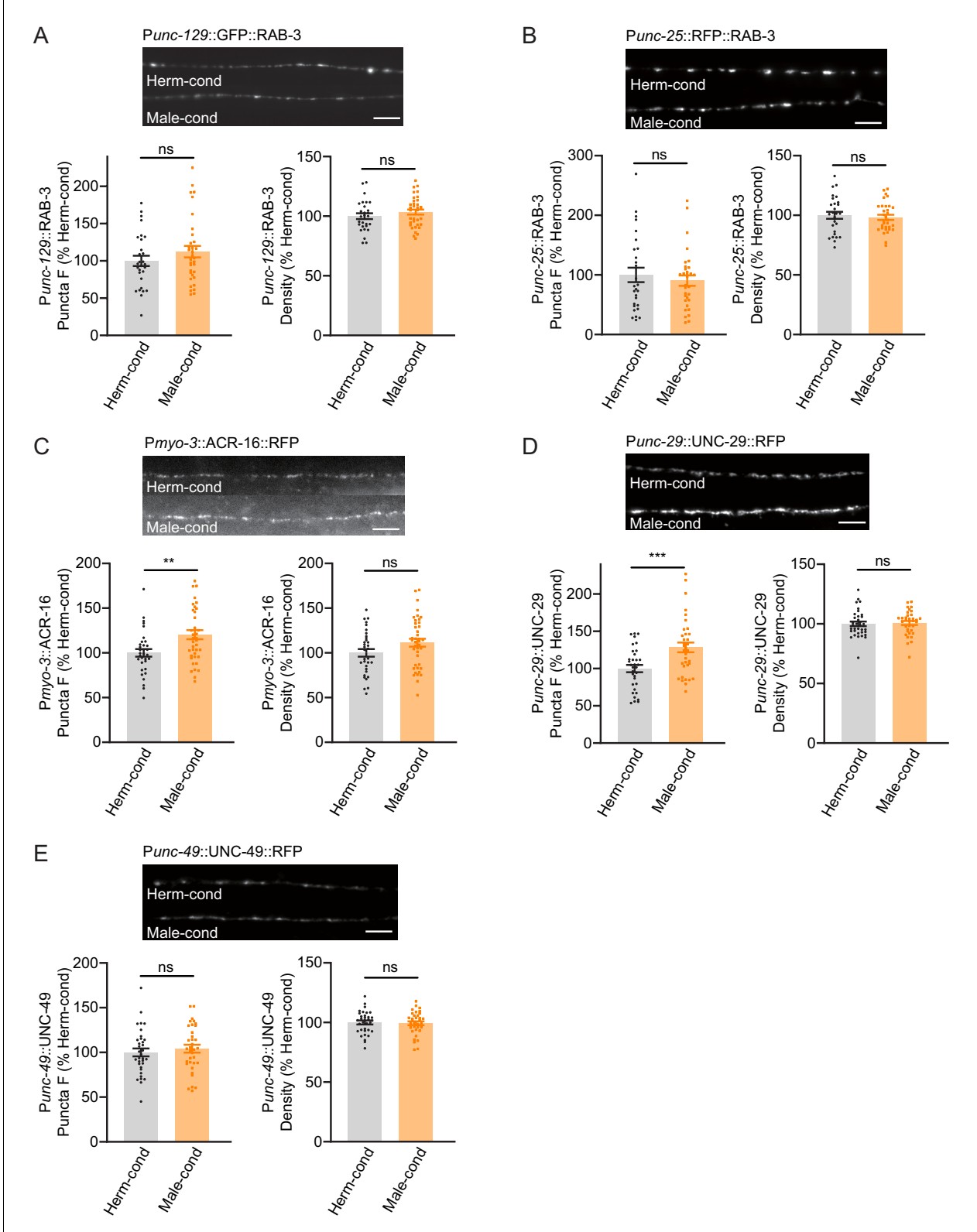

**Figure 5.** Modulator ascarosides promote postsynaptic AchR synaptic localization. (**A**) The puncta fluorescence intensities and densities marked by the excitatory synaptic GFP::RAB-3 (under *unc-129* promoter) in dorsal nerve cord axons were unaltered by modulator ascarosides. Representative images (top panel), mean puncta intensities and puncta density (bottom panel) are shown for hermaphrodites grown in hermaphrodite- or male-conditioned medium. (**B**) The puncta fluorescence intensities and densities marked by the inhibitory synaptic RFP::RAB-3 (under *unc-25* promoter) were unaltered in

*Figure 5 continued on next page*

*Figure 5 continued*

hermaphrodites cultured in male-conditioned medium. Representative images (top), mean puncta intensities and puncta density (bottom) are shown. (C–E) The muscle-specific ACR-16::RFP, UNC-29::RFP, and UNC-49::RFP fluorescence intensities and densities in hermaphrodites cultured in hermaphrodite- and male-conditioned medium. Representative images, mean puncta intensities and puncta density are shown separately. Scale bars, 5 µm. **p<0.005, ***p<0.001, ns not significant, unpaired Student's t-test.

The online version of this article includes the following source data for figure 5:

**Source data 1.**

presynaptic elements in hermaphrodites grown in either hermaphrodite- or male-conditioned medium. To visualize endogenous CaV2 at excitatory or inhibitory synapses separately, we utilized the split GFP complementary system (*Cabantous et al., 2005*; *Kamiyama et al., 2016*). In *C. elegans*, UNC-2 encodes the CaV2 calcium channel α subunit, and we used CRISPR/Cas9 system to insert a sequence coding for seven GFP11 fragments at the C-terminus of UNC-2/CaV2. In parallel, the GFP 1–10 fragment was constitutively expressed in DA and DB cholinergic motor neurons under the control of the *unc-129* promoter or in the GABAergic motor neurons under the control of the *unc-47* promoter (*Figure 6A*). In this way, we were able to monitor the endogenous localization of CaV2 channels at excitatory and inhibitory synapses. To validate the correct subcellular localization, we coexpressed the presynaptic marker UNC-57/Endophilin fused with mCherry. The CaV2-GFP fusion protein formed fluorescent puncta largely co-localized with UNC-57/Endophilin in dorsal cord axons (*Figure 6B,C*, Pearson correlation coefficient 0.7808 ± 0.022 for DA/DB cholinergic motor terminals, and 0.7880 ± 0.0175 for GABAergic motor neuron terminals), confirming that CaV2-splitGFP is localized correctly at presynaptic elements. We further found that UNC-57/endophilin fluorescence intensities and densities were indistinguishable in hermaphrodites from the hermaphrodite- and male-conditioned medium (*Figure 6—figure supplement 1*). This result is consistent with RAB-3-GFP imaging results and support that the presynaptic structure is not altered by male pheromone (*Figure 5A,B*).

Comparison of the CaV2 puncta fluorescence intensities revealed a significant increase at cholinergic synapses of hermaphrodites from male-conditioned medium compared to those from the hermaphrodite-conditioned medium (*Figure 6D*). A slight but notable increase in densities was also observed (*Figure 6D*). In contrast, we detected no significant differences in CaV2 puncta fluorescence intensities and densities at GABAergic synapses (*Figure 6E*).

To further confirm that CaV2 is the synaptic target of modulator ascarosides, we compared the cholinergic synaptic transmission and locomotion velocity in *unc-2* hermaphrodites from male- and hermaphrodite-conditioned medium. The mEPSC rate and locomotion velocity in the *unc-2* mutant were decreased compared to those in the wild type (*Figure 7A,B*), which is correlated with the requirement of CaV2 in mediating presynaptic transmission. Furthermore, we found that the male-specific ascarosides no longer increase mEPSC rates in the *unc-2* hermaphrodites (*Figure 7A–C*). Similarly, the locomotion velocity was not changed in *unc-2* hermaphrodites from male-condition medium compared to those from hermaphrodite-condition medium (*Figure 7D*), which suggests that *unc-2* mutation blocks the effects of the male-specific modulator ascarosides on NMJ synaptic transmission. These findings collectively indicate that the male-specific modulator ascarosides may promote the accumulation of CaV2 calcium channels at excitatory cholinergic synapses, accounting for the potentiated cholinergic synaptic transmission at NMJ.

## Discussion

In this study, we have revealed a novel mechanism through which the male environment modulates the NMJ synaptic transmission, locomotion behavior, and mating efficiency in hermaphrodites. We show that the male environment effects are mediated based on exposure to male-specific pheromones at a specific developmental stage in hermaphrodites (the entire L3–L4 stage). We further demonstrate that hermaphrodite sense and process these male-specific pheromones by AWB chemosensory neurons using the cGMP signaling. At the hermaphrodite NMJ, presynaptic calcium channel localization and postsynaptic acetylcholine receptor clustering are elevated by exposure to male-specific pheromones, resulting in an increased cholinergic synaptic transmission. Our results provide mechanistic details of how environmental factors alter neuronal development and physiology,

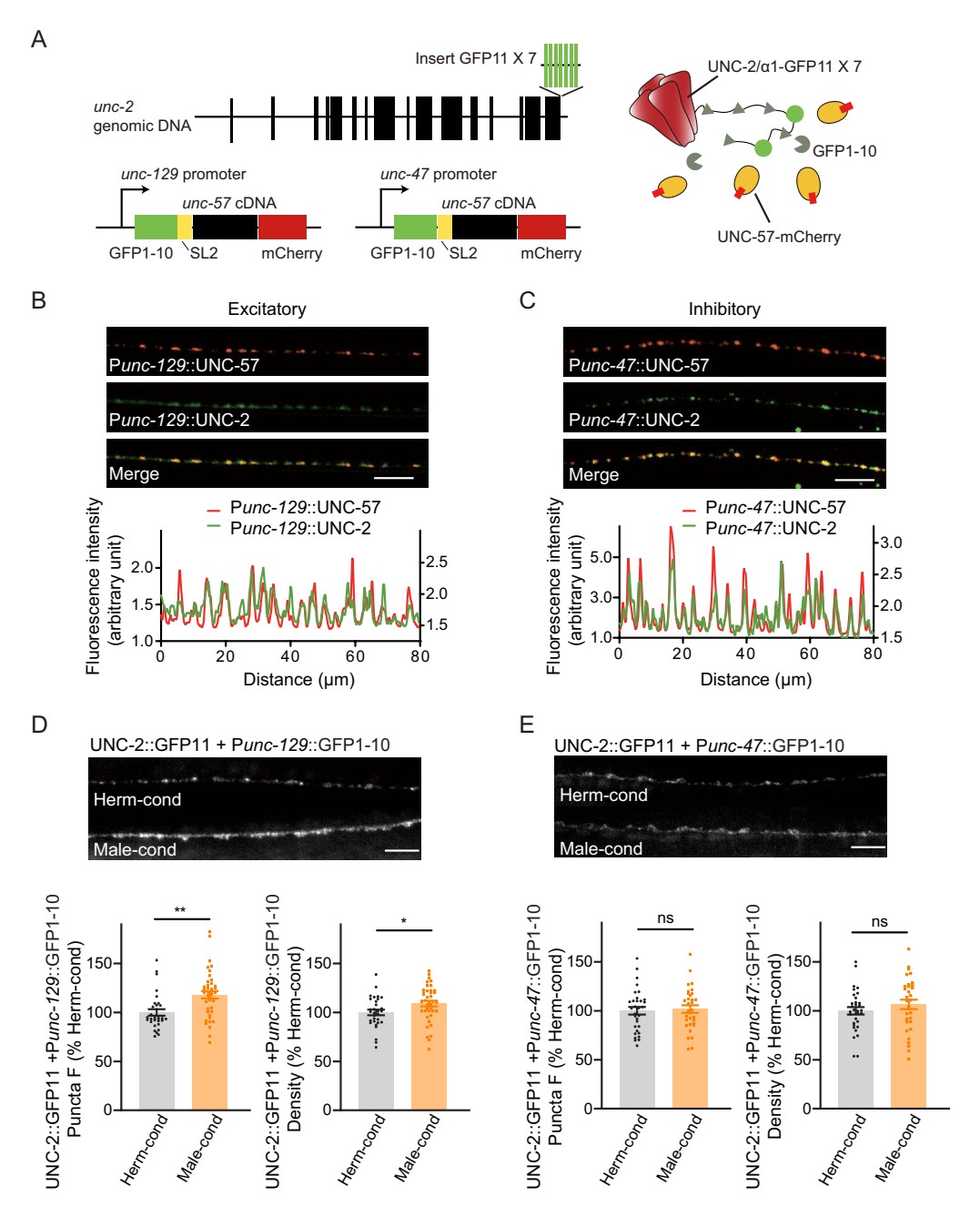

**Figure 6.** Modulator ascarosides increase the abundance of excitatory presynaptic CaV2 calcium channels at the NMJ. (A) Schematic illustration of split GFP experimental design. Seven copies of the splitGFP 11 were inserted into the C-terminal of *unc-2* genomic loci by CRISPR-Cas9 system. The splitGFP1-10 was expressed in B-type cholinergic and GABAergic motor neurons by *unc-129* and *unc-47* promoters. The unc-57-mCherry under the same promoter was separated with splitGFP1-10 by SL2 and was also used as a coexpressed presynaptic marker. (B, C) Presynaptic UNC-2::splitGFP (green) and UNC-57::mCherry (red) were co-localized in the dorsal nerve cord at both excitatory (B) and inhibitory (C) synapses. Representative images (top, scale bar, 10 μm) and linescan curves (bottom) are shown. For linescan curves, the mCherry signals were plotted on the left Y-axis, while the splitGFP signals were plotted on the right. one arbitrary fluorescence intensity unit equals 100 gray value. (D, E) The puncta fluorescence intensities and densities of UNC-2::splitGFP in B-type motor neurons (D) and GABAergic motor neurons (E) of hermaphrodites cultured in hermaphrodite- or male-conditioned medium. Representative images (scale bar, 5 μm), mean puncta intensities, and puncta densities are shown. In D and E, *p<0.05, **p<0.01, ns not significant, unpaired Student's t-test.

The online version of this article includes the following source data and figure supplement(s) for figure 6:

**Source data 1.**

**Figure supplement 1.** Excitatory and inhibitory synapse structures are not affected by the modulator ascarosides.

*Figure 6 continued on next page*

*Figure 6 continued*

**Figure supplement 1—source data 1.**

presenting insights to better understand the associations between dysregulated neurodevelopment and various psychiatric diseases.

## *C. elegans* NMJ as a model to study synaptogenesis

Here, we used the *C. elegans* NMJ as a model to study synaptic transmission, and our work underscore *C. elegans* as a useful model to study synaptic transmission in vivo. The motor circuit of *C. elegans* relies on a precise balance between cholinergic excitation and GABAergic inhibition of body-wall muscles to generates precise locomotion activities (*Richmond and Jorgensen, 1999*). Both our and others' studies have identified mechanisms of synaptogenesis and synaptic transmission that are

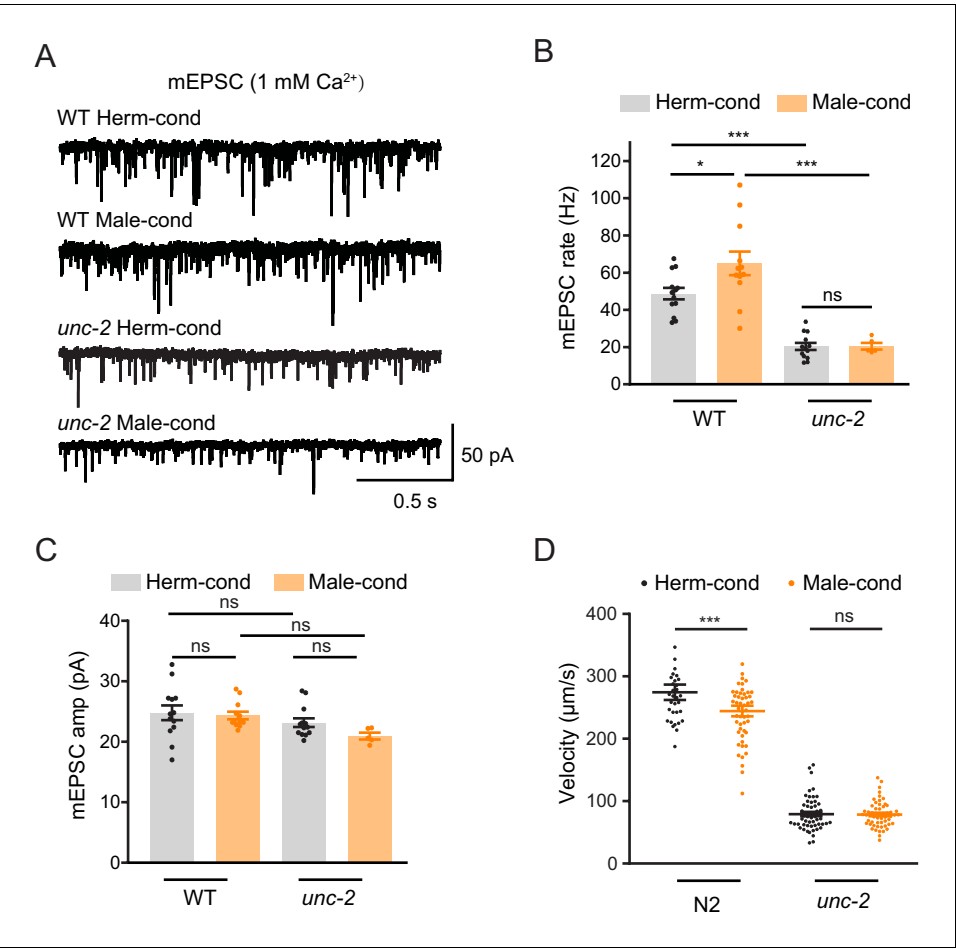

**Figure 7.** CaV2 calcium channel is the synaptic target of the modulator ascarosides. (A–C) Endogenous acetylcholine transmission was assessed by recording mEPSCs from body muscles of wild-type N2 and *unc-2* mutant adult hermaphrodites cultured in hermaphrodite- or male-conditioned medium. Representative mEPSC traces (A), the mean mEPSC rates (B), and the mean mEPSC amplitudes (C) are shown. The data for wild type (N2) is the same as in *Figure 1H–J*. (D) Locomotion behavior analysis of the single wild-type and *unc-2* mutant hermaphrodite in hermaphrodite- and male-conditioned medium. The averaged and individual locomotion velocities were plotted. In (B)–(D), *p<0.05, ***p<0.001, ns not significant, one-way ANOVA with post hoc Dunnett multiple comparisons.

The online version of this article includes the following source data for figure 7:

**Source data 1.**

shared by the *C. elegans* NMJ and the mammalian central nervous system (*Dolphin and Lee, 2020*; *Hata et al., 1993*; *Ogawa et al., 1998*; *Pevsner et al., 1994*; *Richmond et al., 1999*; *Rizo and Südhof, 2012*). In the worm motor circuit and the mammalian brain, acetylcholine is an excitatory neurotransmitter while GABA is an inhibitory neurotransmitter. Moreover, the clustering of acetylcholine receptors and GABA receptors at synapses is observed in *C. elegans* and vertebrates (*Maro et al., 2015*; *Poulopoulos et al., 2009*; *Tong et al., 2015*; *Tu et al., 2015*).

It is also highly notable that many autism-linked synaptic proteins, including Neuroligins and Neurexins, have been shown to function with conserved roles in NMJ synaptogenesis and synaptic transmission (*Hart and Hobert, 2018*; *Hu et al., 2012*; *Kurshan et al., 2018*; *Philbrook et al., 2018*; *Tong et al., 2017*): Neuroligins and Neurexins form trans-synaptic complex and regulate synaptic transmission in both mammalian central nervous system and *C. elegans* NMJ (*Hu et al., 2012*; *Kurshan et al., 2018*; *Tong et al., 2017*). Neuroligins are required for postsynaptic GABA$_A$-receptor clustering and inhibitory synaptic transmission (*Maro et al., 2015*; *Poulopoulos et al., 2009*; *Tong et al., 2015*; *Tu et al., 2015*). While Neurexins undergo ectodomain shedding by ADAM10 protease (*Borcel et al., 2016*; *Tong et al., 2015*), bind to presynaptic CaV2 calcium channel α2δ subunits, and regulate calcium channel activity (*Luo et al., 2020*; *Tong et al., 2017*). Thus, the mechanisms we identified here in the *C. elegans* NMJ may provide new insights into how synaptic transmission is maintained in the mammalian brain.

## Sexual dimorphic modulation on NMJ synaptic transmission

We show that a previously unknown circuit comprised of AWB chemosensory neurons regulates NMJ synaptic transmission in *C. elegans*. Interestingly, the male-enriched pheromones increase the acetylcholine transmission specifically in hermaphrodites but not in males, suggesting sexual dimorphism in the regulation of NMJ synaptic transmission. This could be mediated by sex-specific neuronal circuits that are composed of either sex-specific or sex-shared neurons to process and transmit male pheromone signals to NMJ. A *C. elegans* male has 385 neurons, whereas a hermaphrodite has 302 neurons. The majority of male-specific neurons are localized in the male tail and are involved in the complex mating behaviors. There are several hermaphrodite-specific neurons in the nervous system, including VC and HSN motor neurons, which are mainly required for reproductive functions (*Banerjee and Hallem, 2018*; *Emmons, 2018*; *García and Portman, 2016*). On the other hand, several sex-shared neurons, including motor neurons, AWA, AWC, and ASK chemosensory neurons, DVA mechanosensory neurons, as well as AVA interneurons, could contribute to sex-specific neural circuits by mediating attraction and aversion behaviors (*Banerjee and Hallem, 2018*; *Bayer and Hobert, 2018*; *Cook et al., 2019*; *Fagan et al., 2018*; *Mowrey et al., 2014*; *Narayan et al., 2016*; *Wan et al., 2019*). Our results identified that AWB chemosensory neurons mediate a sexually dimorphic modulation of NMJ synaptic transmission. Further studies will be required to unravel the downstream neural circuits, including interneurons and premotor neurons, that function to process the modulator ascaroside signals to modulate NMJ synaptic transmission. Another possibility for this sexual dimorphic modulation is from sexually dimorphic hormone signaling pathways, such as vasopressin/oxytocin and their receptors (*Garrison et al., 2012*).

Our data show that pheromones modulating hermaphrodite NMJ synaptic transmission are enriched in N2 males. Previous studies have reported various male-specific ascarosides, including ascr#10 and indole containing ascarosides (IC-ascarosides, especially icas#3 and icas#9). However, our data indicate that ascr#10 and indole IC-ascarosides are unlikely the modulator ascarosides. First, ascr#10 levels are comparable in N2 and TR389 males. Second, previous work has established that ASI and ASK sensory neurons are required for hermaphrodites to sense ascr#10 and IC-ascarosides (*Aprison and Ruvinsky, 2017*; *Dong et al., 2016*), whereas we find that ASI and ASK neurons are dispensable for hermaphrodites to sense the modulator ascarosides. In contrast, our UPLC-MS data strongly suggest that the medium-chain β-hydroxylated ascarosides (C13, C14, and C15) may mediate this effect. Although we provided extensive genetic evidence, we have not experimentally confirmed that these specific ascarosides are sufficient to modulate hermaphrodite NMJ synaptic transmission.

Previous studies by Brunet and Murphy labs have shown that male pheromone exposure affects animal health and shortens hermaphrodite life span (*Maures et al., 2014*; *Shi and Murphy, 2014*; *Shi et al., 2017*). Here our data suggest that it might be different mechanisms to modulate longevity and NMJ synaptic transmission. In previous research, they found that exposure of hermaphrodite to

male pheromones at the beginning of their life (day 1) or sexual maturity (day 4) had a similar effect on hermaphrodites' life span. However, we showed that L3–L4 is a critical developmental stage for modulation of hermaphrodite NMJ synaptic transmission by male pheromones (*Figure 2A–C*). Distinct male-specific pheromones may mediate the effects on longevity and NMJ. Further studies should be carried out to identify the specific ascaroside pheromones in males.

Our work demonstrates that early pheromone environment exposure has a long-term effect on synaptic transmission. We suspect that the observed effects may be mediated through endocrine signaling pathways, such as DAF-7/TGF-β and DAF-2/insulin, which are known to drive both epigenomic and transcriptional changes. In this light, recent studies have shown how pheromone exposure can inhibit learning behavior by disrupting the balance between two insulin-like peptides, ins-16 and ins-4 (*Wu et al., 2019*). Further studies are required to characterize whether endocrine system components like insulin signaling molecules are involved in regulating synaptic transmission in response to male-specific ascarosides.

### Presynaptic calcium channels as neuromodulation targets

Our results show that modulator pheromones regulate hermaphrodite NMJ cholinergic transmission by altering the presynaptic localization of calcium channel CaV2 at cholinergic synapses. These results support that CaV2 calcium channels can be viewed as potential targets for environmental modulation of the synaptic transmission. At synapses, CaV2 channels are known to form large signaling complexes in the presynaptic nerve terminal that are responsible for calcium influx and neurotransmitter release (*Dolphin and Lee, 2020*). Numerous studies have verified causal relationships for calcium channel mutations and polymorphisms in neuropsychiatric diseases, including ASD (*Nanou and Catterall, 2018*; *Zamponi, 2016*). Our previous studies identified a synaptic retrograde signal mediated by autism-linked proteins that regulate CaV2 presynaptic localization to alter excitatory synaptic transmission (*Tong et al., 2017*). Here, we present the important evidence that the presynaptic calcium channel CaV2 could also be a target of social interaction modulation to shift the synaptic excitation and inhibition balance. These results support the idea that changes in presynaptic calcium channel localization could be impactful in some forms of ASD.

How might changes in chemosensory neuron activity contribute to presynaptic calcium channel localization? Our results suggest that it is not a general change of CaV2 expression levels because we observed increased presynaptic localization at cholinergic synapses but not at GABAergic synapses. We suspect that the specific synaptic recruitment of CaV2 is somehow potentiated by the modulator ascarosides. Previous studies have suggested that protein interactions are required for cell-surface localization of calcium channels as well as their docking at the active zone. It is therefore possible that pre-synapse specific proteins that are only present at cholinergic synapses may act downstream of the chemosensory circuits to regulate the surface localization of CaV2 channels.

Collectively, our findings reveal a novel mechanism through which pheromones in the environment modulate synaptogenesis and synaptic transmission in the nervous system. Beyond suggesting that calcium channels may be a shared target for both genetic and environmental modulation during development, our study lays a foundation for studies into the signaling and cell-specific functions underlying neurodevelopmental dysfunction.

## Materials and methods

**Key resources table**

| Reagent type (species) or resource | Designation | Source or reference | Identifiers | Additional information |
|---|---|---|---|---|
| Strain, strain background (*C. elegans*) | *C. elegans* strains used in this study are listed in *Supplementary file 1* | | | |
| Strain (*Escherichia coli*) | OP50 | CGC | RRID:WB-STRAIN:OP50 | |
| Sequence-based reagent | Sequence information is listed in *Supplementary file 2* | | | |

*Continued on next page*

*Continued*

| Reagent type (species) or resource | Designation | Source or reference | Identifiers | Additional information |
|---|---|---|---|---|
| Recombinant DNA reagent | pPD49.26 | Addgene (Andrew Fire) | https://www.addgene.org/1686/ | |
| Recombinant DNA reagent | pPD95.75 | Addgene (Andrew Fire) | https://www.addgene.org/1494/ | |
| Recombinant DNA reagent | pCFJ910 | Addgene (Erik Jorgensen) | https://www.addgene.org/44481/ | |
| Recombinant DNA reagent | CZ14527 | Yingchuan B. Qi *Qi et al., 2012* | | Plasmid: Punc-17β::tomm-20N::miniSOG |
| Recombinant DNA reagent | quan0071 | Quan Wen | Xu, T et al., 2018 | Plasmid: Pacr-5::chrimson |
| Recombinant DNA reagent | pSG368 | Shangbang Gao | Gao, S et al., 2018 | Plasmid: GCaMP6f |
| Commercial assay or kit | PureLink HiPure Plasmid Miniprep Kit | Invitrogen | Cat#: K210002 | |
| Commercial assay or kit | QIAprep Spin Miniprep Kit | Qiagen | Cat#: 27106 | |
| Commercial assay or kit | PrimeSTAR Max DNA Polymerase | Takara | Cat#: R045A | |
| Commercial assay or kit | hyPerFUsion high-fidelity DNA polymerase | ApexBio | Cat#: K1032 | |
| Commercial assay or kit | Hieff CLoneTM Plus One Step Cloing Kit | Yeasen | Cat#: 10911ES62 | |
| Chemical compound, drug | Aldicarb | ApexBio | Cat#: B4778 | |
| Chemical compound, drug | All-trans-Retinal | Sigma | Cat#: R2500 | |
| Chemical compound, drug | Geneticin, G418 Sulfate | GOLDBIO | Cat#: G-418–1 | |
| Chemical compound, drug | 2,3-Butanedione monoxime | Sigma | Cat#: B0753 | |
| Chemical compound, drug | Polybead Microspheres 0.10 μm | Polysciences | Cat#: 00876–15 | |
| Chemical compound, drug | Fluospheres carboxylat | Life Science | Cat#: F8813 | |
| Software, algorithm | ImageJ | NIH | https://imagej.nih.gov/ij/download.html | |
| Software, algorithm | Igor pro 6.3 | WaveMetrics | https://www.wavemetrics.com/products/igorpro/igorpro.htm | |
| Software, algorithm | GraphPad Prism 8 | GraphPad | https://www.graphpad.com/scientific-software/prism/ | |
| Software, algorithm | MATLAB | MathWorks | https://www.mathworks.com/products/matlab.html?s_tid=hp_products_matlab | |
| Software, algorithm | MetaMorph | Molecular Devices | https://www.moleculardevices.com/systems/metamorph-research-imaging/metamorph-microscopy-automation-and-image-analysis-software | |
| Software, algorithm | WormLab | MBF Bioscience | https://www.mbfbioscience.com/wormlab | |

## Contact for reagent and resource sharing

Further information and requests for resources and reagents should be directed to and will be fulfilled by the Lead Contact Xia-Jing Tong (tongxj@shanghaitech.edu.cn).

## Experimental model and subject details

### Animals

*C. elegans* were maintained under standard conditions at 20°C on plates made from nematode growth medium (NGM). *E. coli* OP50 was used as a food source for all experiments except where HB101 *E. coli* was utilized for electrophysiology study. A description of all alleles can be found at http://www.wormbase.org/#012-34-5. Animals were obtained from Bristol variety N2 strain unless specially indicated. Transgenic animals were prepared by microinjection, and integrated transgenes were isolated following UV irradiation or by miniMos insertion.

### Plasmids

All worm expression vectors were modified versions of pPD49.26 (A. Fire) or miniMos vector pCFJ910. Standard methods and procedures were utilized for all of the plasmids. A 3.1 kb *ceh-36* promoter was used for expression in ASE and AWC chemosensory neurons. A 3 kb *odr-10* promoter was used for expression in AWA chemosensory neurons. A 3 kb *str-2* promoter was used for expression in AWC chemosensory neurons. A 3 kb *str-1* promoter was used for expression in AWB chemosensory neurons. A 3 kb *srb-6* promoter was used for expression in ADF, ADL, and ASH chemosensory neurons. A 3 kb *gpa-4* promoter was used for expression in ASI chemosensory neurons. A 3 kb *gcy-15* promoter was used for expression in ASG chemosensory neurons. A 3.9 kb *sra-7* promoter was used for expression in ASK chemosensory neurons. A 4.1 kb *flp-21* promoter was used for expression in the majority of the chemosensory neurons. A 4.3 kb *acr-5* promoter was used for expression in DB and VB motor neurons. A 2.4 kb *myo-3* myosin promoter was used for expression in body muscles. For rescue experiments, *TAX-2* (F36F2.5.1), *ODR-3* (C34D1.3.1), and *ACR-16* (F25G6.3) were amplified from the N2 cDNA library using gene-specific primers.

## Generation of single-copy insertion allele by homologous recombination

The xjSI0002 allele encoding RFP-tagged *ACR-16* minigene under the muscle-specific *myo-3* promoter was generated by miniMOS (*Frøkjær-Jensen et al., 2014*). The RFP sequence was inserted between the third and the fourth transmembrane segment of ACR-16.

## Aldicarb assay

The aldicarb assay was performed as previously described (*Vashlishan et al., 2008*). Aldicarb (Apex-Bio) was dissolved in ethyl alcohol and added to NGM at a final concentration of 1.4 mM (Testing hermaphrodites) or 0.5 mM (Testing males). These plates (35 mm) were seeded with 75 µl OP50 and allowed to dry overnight before use. More than 20 animals at the young adult stage (otherwise indicated) were picked on an aldicarb plate for aldicarb assay. Animals were scored as paralyzed when they did not respond to the platinum wire prodding. The paralyzed animals were counted every 10 or 15 min. At least three double-blind replicates for each group were tested.

## Preparation of conditioned media

Hermaphrodite- and male-secreted metabolites were collected according to a previous publication (*Srinivasan et al., 2008*). Synchronized *C. elegans* (WT [N2], *him-5* [N2], WT [TR389], *him-5* [TR389], *daf-22* [N2], and *daf-22; him-5* [N2]) with a density of 10,000 worms/plates (90 mm, three plates) were grown on the nematode growth media (NGM) agarose (seeded with *E. coli* strain OP50) at 20° C. There were 43.07 ± 0.77%, 39.26 ± 1.55%, and 37.29 ± 1.28% males in *him-5* (N2), *daf-22; him-5* (N2), and *him-5* (TR389) strains, respectively. After worms reached the young adult stages, they were collected and washed three times with M9 buffer to remove bacteria. To further remove the bacteria in the gut, the worms were then placed in M9 buffer in a shaker (150 rpm) at 20°C for 30 min and rinsed three times with ddH$_2$O. Subsequently, worm-secreted metabolites were collected by incubating the worms in ddH$_2$O in a shaker (150 rpm) for 3 hr with a density of 30,000 worms/ml. Afterward, the worms were removed by settling on ice for 5 min. The metabolites were filtered

through 0.22 µm filters, aliquoted, and stored at −80℃. For conditioned medium preparation, 10 µl metabolites mixed with 90 µl OP50 *E. coli* were spread on a 35 mm NGM plate. Plates were allowed to dry overnight before use.

## Calcium imaging

Muscle calcium responses were measured by detecting fluorescence intensity changes of GCaMP3. *C. elegans* expressing GCaMP3 in the body-wall muscle (P*myo-3*::GCaMP3) and Chrimson in the DB and VB neurons(P*acr-5::*Chrimson) were used for calcium imaging experiments. Young adult animals fed with 1.6 mM ATR (in 100 µl *E. coli* OP 50) were immobilized on 10% agarose pads by polystyrene microbeads. Fluorescence images were captured using a Nikon 60 × 1.4 NA objective on a Nikon spinning-disk confocal system (Yokogawa CSU-W1) at 10 ms per frame. We used wide-field illumination with a nominal wavelength at 640 nm for Chrimson activation. The GCaMP signals were captured using 488 nm laser excitation and were analyzed by ImageJ software.

Calcium responses in the soma of AWB sensory neurons were measured by detecting fluorescence intensity changes of GCaMP6f. A home-made microfluidic device was used for calcium imaging as previously described (*Liu et al., 2019*; *Zou et al., 2018*). Briefly, a young adult animal was rinsed by M9 buffer and loaded into a home-made microfluidic device with its nose exposed to buffer under laminar flow. Diluted metabolites of N2 hermaphrodite and *him-5* mutants was delivered using a programmable automatic drug feeding equipment (MPS-2, InBio Life Science Instrument Co. Ltd, Wuhan, China). For $Ca^{2+}$ fluorescence imaging in AWB, the neurons were exposed under fluorescent excitation light for 30 s before recording, to eliminate the light-evoked calcium transients. During the recording process, for the first 5 s, we gave the M9 solution and then switched hermaphrodite excretome or male excretome for 30 s, removing extract liquid from 35 s and washing for 25 s. The AWB neurons were imaged with a 60× objective (Olympus) and EMCCD camera (Andor iXonEM) at 100 ms/frame. The imaging sequences were subsequently analyzed using Image-Pro Plus6 (Media Cybernetics, Inc, Rockville, MD).

## Adult locomotion analysis

To analyze adult locomotor behavior, young adult worms were washed with M9 buffer before transferred to the unseeded NGM plate and allowed to recover for 5 min. Animal locomotion was recorded at a rate of 10 frames per second for 1 min. The mean body-bend amplitude and crawling locomotion velocity were analyzed by WormLab. All the assays were done at 25℃.

## Mating behaviors

Mating efficiency was assessed as previously described (*Yin et al., 2017*). Briefly, two young adult stage hermaphrodites from male- or hermaphrodite-conditioned medium were cultured with two young-adult males for 24 hr. Successful mating was scored when more than three male progenies were generated in the mating plate. Mating efficiency was obtained by calculating the percentage of successful mating in more than 15 plates.

## Liquid culture and mass spectrometric analysis

The crude pheromone extracts were prepared according to a previously published protocol (*Zhang et al., 2013*). N2 wild-type, *him-5* mutant, *daf-22; him-5* mutant, or TR389 *him-5* mutant worms were grown for two generations on 60 mm NGM plate seeded with *E. coli* OP50 bacteria. Worms from four plates were washed by M9 buffer and cultured in 50 ml S complete medium (100 mM NaCl, 50 mM $KPO_4$, 3 mM $CaCl_2$, 3 mM $MgSO_4$, 5 µg/ml cholesterol, 10 mM potassium citrate, 50 µM disodium EDTA, 25 µM $FeSO_4$, 10 µM $MnCl_2$, 10 µM $ZnSO_4$, 1 µM $CuSO_4$) at 20℃ and 200 rpm. The animals were cultured with shaking at 200 rpm for 7 days (around 30,000 worms/50 ml). 25× Concentrated *E. coli* OP50 bacteria were supplemented every day (0.3 ml for day 1, 1 ml for days 2–5, and 2 ml for days 6–7). After 7 days, the supernatant medium containing metabolites was collected by centrifugation (3000 g, 10 min). Then the supernatant media were frozen at −80℃, lyophilized, and extracted with methanol for UPLC-MS analysis. UPLC-MS was performed using a Sciex TripleTOF 6600 system. Chromatographic separations were achieved using a Waters Acquity UPLC BEH C18 column (1.7 µm, 2.1 × 10 mm), with a flow rate of 0.4 ml/min. Data acquisition and processing were performed by Analyst and Peakview (Sciex).

## Genetic ablation with miniSOG

The genetically encoded photosensitizer, miniSOG, was used to ablate specific neurons as previously described (*Qi et al., 2012*). Synchronized Late L1 larva animals (24 hr after egg hatching at 20℃) expressing miniSOG under specific promoters were exposed to wide-field blue light (460 nm) at an intensity of 57 mW/cm$^2$ for 30 min, and then animals were grown in the 20℃ incubator before experiments. The ablation efficiency was measured by comparing mCherry fluorescent signal with and without blue light stimulation.

## Optogenetic activation of chemosensory neurons

To prepare the plates for optogenetic activation of neurons, 1.6 mM of all-trans-retinal (ATR, 100 mM dissolved in ethanol) or ethanol (control) was mixed with OP50 *E. coli* culture and spotted on 35 mm NGM plates. Plates were allowed to dry for 24 hr before usage. Transgenic worms with channelrhodopsin variant CHIEF expressed in ASJ/ASI, AWA, AWB, or AWC chemosensory neurons were grown overnight on the NGM plates. Animals at L4 larval stages received 100 ms pulse stimulation of blue light (460 nm wavelength, 2.4 mW/mm$^2$ power) for 10 min (five times) until the animals entered the adult stage.

## Fluorescent microscopy imaging

For quantitative analysis of fluorescence intensities and densities, images were captured using a 100× (NA = 1.4) objective on an Olympus microscope (BX53). Young adult worms were immobilized with 30 µg/µl 2,3-butanedione monoxime. The maximum intensity of dorsal cord projections of Z-series stacks was obtained by Metamorph software (Molecular Devices). Line scans were analyzed in Igor Pro (WaveMetrics) using a custom script (*Dittman and Kaplan, 2006*). The mean fluorescence intensities of reference FluoSphere microspheres (0.5 µm, ThermoFisher Scientific) were measured during each experiment controlled for changes in illumination intensities. To assess the synaptic accumulation of fluorescent proteins, we used the calculation of $\Delta F/F$ as $(F_{puncta} - F_{axon})/F_{axon}$. And we also counted the density of fluorescent puncta.

## Electrophysiology

Electrophysiology was conducted on dissected *C. elegans* as previously described (*Hu et al., 2012*). Worms were superfused in an extracellular solution containing 127 mM NaCl, 5 mM KCl, 26 mM NaHCO$_3$, 1.25 mM NaH$_2$PO$_4$, 20 mM glucose, 1 mM CaCl$_2$, and 4 mM MgCl$_2$, bubbled with 5% CO$_2$, 95% O$_2$ at 22℃. Whole-cell recordings were carried out at −60 mV for mEPSCs, and 0 mV for mIPSCs. The internal solution contained 105 mM CH$_3$O$_3$SCs, 10 mM CsCl, 15 mM CsF, 4 mM MgCl$_2$, 5 mM EGTA, 0.25 mM CaCl$_2$, 10 mM HEPES, and 4 mM Na$_2$ATP. The solution was adjusted to pH 7.2 using CsOH.

## Statistics

All data were reported as mean ± SEM (standard error of the mean). Statistical analyses were performed using GraphPad Prism (version 8). We calculated p-values by two-way ANOVA (*Figures 1B, D* and *2D–F*, *Figure 2—figure supplement 1A–E*, *Figure 2—figure supplement 2*, *Figure 2—figure supplement 4*, *Figure 4—figure supplement 2A–G*, *Figure 4—figure supplement 3A–C*), two-way ANOVA with post hoc Sidak multiple comparisons (*Figures 2B–C, G, 3B–C, F–G, 4C, E, and I–J*, *Figure 3—figure supplement 1A–B*), one-way ANOVA with post hoc Dunnett multiple comparisons (*Figures 2H–I, 3E, 4D, and 7B–D*, *Figure 3—figure supplement 2A–C*), and unpaired Student's t-test (*Figures 1F–G, I–J, L–M, 4H, 5A–E, and 6D–E*, *Figure 1—figure supplement 1B*, *Figure 2—figure supplement 3A and C*, *Figure 6—figure supplement 1A,B*). In all figures, p-values are denoted as *<0.05, **<0.01, ***<0.001.

## Acknowledgements

We thank the *C. elegans* Genetics Stock Center, National BioResource Project (NBRP), Jean-Louis Bessereau, Yingchuan Billy Qi, and Quan Wen for sharing strains and reagents. We also thank the Molecular Imaging Core Facility (MICF), Biological Mass Spectrometry Core Facility (BMSCF) at School of Life Science and Technology, ShanghaiTech University for help in calcium imaging and

mass spectrometric analysis. This work was supported by the Basic Research Project from the Science and Technology Commission of Shanghai Municipality (19JC1414100 to X-JT), National Natural Science Foundation of China (31741054 to X-JT and 31771154 to QL), Shanghai Pujiang Program (18PJ1407600 to X-JT), Shanghai Brain-Intelligence Project from the Science and Technology Commission of Shanghai Municipality (18JC1420302), Program for Special Appointment at Shanghai Institutions of Higher Learning (QD2018017 to QL), Innovative research team of high-level local universities in Shanghai, National Institute of Neurological Disorder and Stroke (NS32196 to JMK), National Institutes of Health research grant (NEI 1R21EY029450-01 to JMK and ZH), and National Health and Medical Research Council (APP1122351 to ZH).

## Additional information

### Funding

| Funder | Grant reference number | Author |
|---|---|---|
| Science and Technology Commission of Shanghai Municipality | 19JC1414100 | Xia-Jing Tong |
| Science and Technology Commission of Shanghai Municipality | 18PJ1407600 | Xia-Jing Tong |
| National Natural Science Foundation of China | 31741054 | Xia-Jing Tong |
| Science and Technology Commission of Shanghai Municipality | 18JC1420302 | Qian Li |
| Shanghai Municipal Education Commission | QD2018017 | Qian Li |
| National Natural Science Foundation of China | 31771154 | Qian Li |
| National Institute of Neurological Disorders and Stroke | NS32196 | Joshua M Kaplan |
| National Institutes of Health | NEI 1R21EY029450-01 | Zhitao Hu<br>Joshua M Kaplan |
| National Health and Medical Research Council | APP1122351 | Zhitao Hu |

The funders had no role in study design, data collection and interpretation, or the decision to submit the work for publication.

### Author contributions

Kang-Ying Qian, Conceptualization, Resources, Data curation, Formal analysis, Validation, Investigation, Visualization, Methodology, Writing - original draft, Writing - review and editing; Wan-Xin Zeng, Lili Chen, Fu-min Tian, Resources, Data curation, Formal analysis, Validation, Investigation, Visualization, Methodology; Yue Hao, Resources, Data curation, Formal analysis, Investigation, Visualization, Methodology; Xian-Ting Zeng, Resources, Validation, Methodology; Haowen Liu, Lei Li, Resources, Data curation, Formal analysis, Validation, Investigation, Methodology; Cindy Chang, Qi Hall, Resources; Chun-Xue Song, Resources, Data curation, Formal analysis, Investigation, Methodology; Shangbang Gao, Data curation, Formal analysis, Supervision, Investigation, Visualization, Methodology; Zhitao Hu, Data curation, Formal analysis, Supervision, Funding acquisition, Validation, Methodology; Joshua M Kaplan, Resources, Supervision; Qian Li, Conceptualization, Formal analysis, Supervision, Funding acquisition, Validation, Investigation, Visualization, Writing - original draft, Project administration, Writing - review and editing; Xia-Jing Tong, Conceptualization, Formal analysis, Supervision, Funding acquisition, Validation, Investigation, Visualization, Methodology, Writing - original draft, Project administration, Writing - review and editing

## Author ORCIDs

Kang-Ying Qian  https://orcid.org/0000-0001-7791-3770
Shangbang Gao  http://orcid.org/0000-0001-5431-4628
Zhitao Hu  https://orcid.org/0000-0002-2948-3339
Joshua M Kaplan  https://orcid.org/0000-0001-7418-7179
Qian Li  https://orcid.org/0000-0003-1300-3377
Xia-Jing Tong  https://orcid.org/0000-0001-5634-1136

## Decision letter and Author response

Decision letter https://doi.org/10.7554/eLife.67170.sa1
Author response https://doi.org/10.7554/eLife.67170.sa2

## Additional files

### Supplementary files

- Supplementary file 1. C. elegans strains used in this study.
- Supplementary file 2. Sequence information.
- Transparent reporting form

### Data availability

All data generated or analysed during this study are included in the manuscript and supporting files. Source data files have been provided. For other information (such as primers), we have included them in the methods.

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
