## [Decision Letter]

**Acceptance summary:**

Your paper reports the identification of a fascinating and surprising effect of male pheromones on the behavior of *C. elegans* hermaphrodites. With a powerful combination of behavioral assays, electrophysiology, cell biology, and chemical analysis, your experiments show that developmental exposure of hermaphrodites to male-derived pheromones alters the physiology of the hermaphrodite neuromuscular junction and provide significant insight into the mechanism by which this occurs. Your work adds an important new dimension to our understanding of the effects of pheromone signals in *C. elegans*.

**Decision letter after peer review:**

[Editors’ note: the authors submitted for reconsideration following the decision after peer review. What follows is the decision letter after the first round of review.]

Thank you for submitting your work entitled "Male pheromones modulate synaptic transmission at the *C. elegans* neuromuscular junction in a sexually dimorphic manner" for consideration by *eLife*. Your article has been reviewed by 3 peer reviewers, including Douglas Portman (Reviewer #1), and the evaluation has been overseen by a Senior Editor. The following individual involved in review of your submission has agreed to reveal their identity: Kyuhyung Kim (Reviewer #3).

Our decision has been reached after consultation between the reviewers. Based on these discussions and the individual reviews below, we regret to inform you that your work will not be considered further for publication in *eLife* in its current form.

As you will see, all three reviewers feel that your findings, which suggest that male pheromones can influence the physiology of the hermaphrodite neuromuscular junction, are potentially quite interesting and exciting. However, there are substantial concerns about the work as it stands. In particular, it is not clear that the effects of conditioned media you report truly result from exposure to male pheromones.

We would be happy to consider a significantly revised and improved paper as a new submission. Such a paper would need to address the following issues:

1. Provide evidence that the conditioned media was prepared in a standardized way that eliminates concerns about the potential artifacts. Otherwise, all experiments with conditioned media would need to be redone.

2. Provide a more thorough analysis of changes in synaptic architecture by testing additional markers (such as cla-1) and quantitating puncta number/size.

3. Carry out calcium imaging to determine whether AWB detects male pheromones.

4. Provide additional details on statistical analysis as well as missing controls as described below.

5. If possible, testing synthetic ascarosides would significantly improve the paper, though we understand that this may be difficult or impossible to achieve.

Reviewer #1:

This interesting manuscript from Qian reports that *C. elegans* males may sex-specifically release compounds that alter NMJ physiology and locomotion in the hermaphrodite. Using a powerful combination of electrophysiology, behavioral analysis, optogenetics, genetic ablations, and imaging, the authors provide evidence that media conditioned by *C. elegans* males increases the E/I balance. Their results suggest that exposure during a critical period in late larval development is required for this effect; that exposure to conditioned media increases mEPSC frequency at the NMJ; that the effect is likely due to ascaroside pheromones; that these pheromones may be detected by the AWB neurons; that exposure to conditioned media changes pre- and post-synaptic CaV2 and AChR abundance, respectively; and that the CaV2 channel unc-2 is required for the physiological and behavioral effects of conditioned medium. A number of important issues remain unclear, such as the molecular identity of the relevant pheromone(s), whether these molecules are directly sensed by AWB, and whether changes at the NMJ are direct results of pheromone detection or are instead indirect changes that occur secondary to other changes in physiology that male pheromones can elicit in hermaphrodites. Nevertheless, the paper reports interesting and surprising findings that provide new insight into social and sexual interactions in *C. elegans* and the roles of pheromones in modulating physiology and behavior.

1. A very significant concern (perhaps a fatal flaw) is that essential information about the methods and details of the experiments are lacking. Without these, it becomes impossible to assess the validity of the experimental design and interpretation. In particular, it is unclear how conditioned media (CM) was prepared and what method was used to expose animals to it. If the liquid culture method on p. 29 was used to prepare CM, this likely makes much of the data in the paper uninterpretable. Because these cultures are grown for 7 days, they likely differ greatly in the number of animals as well as their population structure (relative numbers of different larval stages, dauers, and adults of various ages). These cultures are also susceptible to bacterial and fungal contamination and cause animals significant stress. As such, direct comparisons of the activity of CM from N2, him-5, and daf-22 CM is highly problematic, as any of the above could account for differences in its activity. Without knowing that CM was prepared from a defined number of animals of defined ages, it's impossible to draw conclusions about the basis for the differential effects of CM. The absence of any experiments with synthetic ascarosides compounds these concerns.

2. Statistics and data presentation. Not enough information is provided to understand how aldicarb sensitivity was statistically evaluated (it's particularly hard to understand the apparent significant differences in the results shown in 4C). Further, scatter plots need to be provided in place of the histograms shown in Figures1, 2, 5, 6, 7 and supplementary.

Reviewer #2:

The authors identify a fascinating and novel phenotype in hermaphrodite *C. elegans*, where synaptic transmission at the neuromuscular junction (measured using aldicarb response) and locomotion behavior are altered by the presence of male *C. elegans*. The authors demonstrate that this change is due to pheromones released from males, and that exposure to males during development is temporally dependent. The authors also do a preliminary dissection of what pheromones mediate the phenotype, the sensory circuitry required for this phenotype, and some mechanistic analysis at the NMJ. While the work is appropriate in scope and would be exciting for the readership of *eLife*, a number of the major conclusions of the manuscript are hard to evaluate due to lack of important experimental and statistical methodology and a number of crucial experiments lacking rigorous controls.

1. Detailed descriptions of experimental methodology and statistical analyses are lacking for some of the most crucial experiments and thus a number of the major findings are not able to be evaluated in a rigorous manner.

Aldicarb assays – specifics on aldicarb plate is needed- "Briefly,1 mM aldicarb was added to the NGM plate". Variability between aldicarb plates is common and often mentioned in *C. elegans* literature, thus specific details are needed to evaluate how plates were made and maintained, and controls need to be included for experiments in order to evaluate and compare results. Specific details on what values were compared using t-test for aldicarb assays is unclear and needs to be described (eg. on a single time point?).

Conditioned media experiments – This reviewer could find no method descriptions for conditioned media experiments- How was male and hermaphrodite conditioned media made and administered to worms? This is critical for many of the experiments and is especially important to know for male-conditioned media generated from non-him mutant strains (daf-22, N2, and TR389).

2. Appropriate controls are missing in many experiments or are shown in separate graphs making it hard for reviewers/readers to assess phenotypes compared to positive and negative controls, or to evaluate if mutant or transgenic worms have phenotypes in control conditions. Examples include Figure 2D-F, Figure 3C, Figure 4C-L, Figure 7A-E

3. The results in Figure 2 demonstrate temporal dependence, however, to conclude that L4 is the important stage ("exposure of the male environment in the entire L4 stage is critical for modulation of the NMJ synaptic transmission in hermaphrodites" "male environment exposure at a critical period (the L4 stage) is required for the modulation of hermaphrodites NMJ cholinergic synaptic transmission.") would require exposure to male-conditioned media during only the L4 stage, as well as after larval development (adult only exposure to male-conditioned media). Some effects shown here could be explained by duration of exposure. An ANOVA should be performed to allow readers to compare between all timepoints for both herm and male conditioned media conditions.

4. For the figure and section "Excitatory postsynaptic receptor clustering is increased in hermaphrodites exposed to the male environment" – fluorescence intensity is not a good measure of synaptic structure or clustering. Additional markers beyond rab-3 (e.g. cla-1) are needed to evaluate presynaptic morphology and ideally pre and post synaptic clustering experiments would include quantification of the number or size of synaptic puncta.

5. The analysis of ascarosides in male vs herm conditioned media identifies a subset of "modulator ascarosides" and the genetic experiments suggest these may be responsible for the phenotype of male conditioned media. However, without testing the direct effect of the ascarosides on the phenotype, the wording of subsequent results and experiments throughout needs to be toned down ("Thus, the modulator ascarosides cause increased postsynaptic receptor clustering at excitatory synapses in hermaphrodites.", "These findings collectively indicate that the modulator ascarosides specifically promote accumulation of CaV2 calcium channel at excitatory cholinergic synapses, and the increased CaV2 calcium channel abundance account for the potentiated cholinergic synaptic transmission at NMJ.", etc). This is mentioned by the authors in discussion, but needs to be applied to the Results sections ("Although we provided extensive genetic evidence, we have not experimentally confirmed that these specific ascarosides are sufficient to modulate hermaphrodite NMJ synaptic transmission.").

Reviewer #3:

In this manuscript, Qian et al., describes mechanisms how *C. elegans* male pheromones mediate sexually dimorphic impact on synaptic transmission at the NMJ and thus locomotive behaviors. First, the authors found that the male environment, which represents a male-conditioned medium, increases NMJ excitation and inhibition balance of hermaphrodites by specifically increasing cholinergic transmission but not GABAergic transmission, resulting in decreased locomotive activities in hermaphrodites. Then they identified a critical period (the L4 larval stage) required for modulation of the male-conditioned medium on NMJ synaptic transmission in hermaphrodites. Second, by performing candidate mutant test and LC-MS analysis, they identified chemical components in the male-conditioned medium, which are the medium-chain β-hydroxylated ascaroside pheromones. These pheromones appeared to be sensed by the AWB chemosensory neurons and transmitted via cGMP signaling. Finally, they showed that hermaphrodites exposed to the male-conditioned medium exhibit increased postsynaptic nicotinic acetylcholine receptor clustering as well as increased presynaptic CaV2 calcium channel localization at NMJ.

Overall, I think this is a very interesting story, with novel findings and implications about the pheromone-mediated behavioral plasticity in this important model organism. There are a couple of places where I think the authors could fill gaps in the story that would make it more complete and logically compelling.

Several major points that need to addressed and improved include;

1. Authors should test whether chemically synthesized ascarosides, including at least C13 and C14, individually and in a mixture, recapitulate effects of the male-conditioned medium on aldicarb assay as well as electrophysiology experiment. Without these data, it is hard to assure that these pheromones mediate it.

2. Authors claimed that the AWB chemosensory neurons detect these pheromones and mediate effects of male-conditioned medium mainly by performing cell-ablation and mutant studies. Nowadays, it is quite common to do Ca^2+^ imaging experiments using genetically encoded Ca^2+^ sensors, which make a solid conclusion that AWB neurons detect these pheromones.

3. Authors have not addressed or discussed at all why male affects hermaphrodite locomotion. Is mating efficiency or physiology such as egg-laying of hermaphrodites altered in the male environment?

[Editors’ note: further revisions were suggested prior to acceptance, as described below.]

Thank you for resubmitting your work entitled "Male pheromones modulate synaptic transmission at the *C. elegans* neuromuscular junction in a sexually dimorphic manner" for further consideration by *eLife*. Your revised article has been evaluated by Piali Sengupta (Senior Editor) and a Reviewing Editor (Doug Portman), and also seen by previous reviewers.

The reviewers feel that your manuscript has been significantly improved by the inclusion of new data and clarification of some of technical points. However, there are several additional points that need to be addressed before acceptance, as outlined by Reviewer #2. In particular, please be sure to address each of the four major comments noted by Reviewer #2 in your resubmission. None of these require new experiments, but instead can be addressed by carrying out additional statistical tests and/or editing the text.

Reviewer #2:

This revised manuscript is very exciting, and will be of interest to the readership of *eLife*. The authors have included many revisions and new experiments that make the revised manuscript rise to the level of publication in *eLife* in terms of rigor and novelty. There are a few changes that would further improve clarity on some experiments.

1. While the authors have included much more detail on statistical analysis and methods, it should be clearly stated exactly what comparison the p-values represent for aldicarb assays (eg. Figure 1b- all time points? One timepoint?).

2. Figure 2C. Please provide p-value comparing L4 out vs adult out.

3. The experiment in figure 4E appears to show the addition of ATR lowers aldicarb sensitivity and that light exposure rescues this. P-values comparing all conditions should be included. While ATR can impact behavior and does not necessarily take away from the finding, the wording should be altered to reflect if there are differences between the non-ATR controls and the ATR and ATR + light condition.

Pg 15 line 21 "Hermaphrodites with activated AWB neurons during the L4 stage showed higher sensitivity to aldicarb than controls without blue light activation (Figure 4E)."

This statement suggests that the final bar is higher than all control bars, which is not shown statistically.

4. In Figure 4D, it appears that the velocity of worms with AWB ablation is reduced in both the herm- and male-cond compared with no light controls. Please provide p-values comparing all conditions, and if the herm-cond AWB ablation is reduced compared with no light herm-cond, please alter the wording of the statement below to reflect this data.

"Pg 15 line 14. Our data showed that AWB neurons ablation blocked the decreased velocity by the modulator ascarosides (Figure 4D)."

Reviewer #3:

Although they could not perform the experiment with synthetic chemicals (this is very challenging and I accepted their explanation), the authors have addressed most of my concerns and improved the manuscript considerably. I am now fully supportive of publishing this manuscript.

---

## [Author Response]

[Editors’ note: the authors resubmitted a revised version of the paper for consideration. What follows is the authors’ response to the first round of review.]

As you will see, all three reviewers feel that your findings, which suggest that male pheromones can influence the physiology of the hermaphrodite neuromuscular junction, are potentially quite interesting and exciting. However, there are substantial concerns about the work as it stands. In particular, it is not clear that the effects of conditioned media you report truly result from exposure to male pheromones.We would be happy to consider a significantly revised and improved paper as a new submission. Such a paper would need to address the following issues:1. Provide evidence that the conditioned media was prepared in a standardized way that eliminates concerns about the potential artifacts. Otherwise, all experiments with conditioned media would need to be redone.

We apologize for not including the information in the previous manuscript. We have added the detailed description of the conditioned media preparation in Figure 1C and Experimental model and subject details.

2. Provide a more thorough analysis of changes in synaptic architecture by testing additional markers (such as cla-1) and quantitating puncta number/size.

According to the reviewer’s suggestion, we performed experiments to measure the fluorescence intensities and densities of UNC-57/Endophilin as another synaptic marker. We found that UNC57/Endophilin fluorescence intensities and densities were indistinguishable in hermaphrodites from the hermaphrodite- and male-conditioned medium (Supplementary Figure 11). This result is consistent with RAB-3-GFP imaging results, and support that the presynaptic structure is not altered by male pheromone (Figure 5A-B). We included the data in Supplementary Figure 11.

We also quantified puncta density of RAB-3, cholinergic receptors (ACR-16, UNC-29), and GABAergic receptors (UNC-49) in hermaphrodites from hermaphrodite- and male-conditioned medium. We included the densities results in Figure 5.

3. Carry out calcium imaging to determine whether AWB detects male pheromones.

We thank reviewer#3 for raising this critical question. We have conducted calcium imaging to monitor intracellular Ca^2+^ dynamics upon male pheromone stimulation in AWB soma. We found that the AWB neurons elicited a rapid and robust calcium transient response to the male excretomes, but not hermaphrodite excretomes. We included this data in Figure 4F-H and revised our manuscript on page 16 line 3-8.

4. Provide additional details on statistical analysis as well as missing controls as described below.

Statistics

All data were reported as mean ± SEM (standard error of the mean). Statistical analyses were performed using GraphPad Prism (version 8). We calculated p values by two-way ANOVA (Figure 1B, 1D, 2D-F, Supplementary Figure 2A-E, 3, 5, 9A-G, 10A-C), two-way ANOVA with post-hoc Sidak multiple comparisons (Figure 2B-C, 2G, 3B-C, 3F-G, 4C, 4E, 4I-J, Supplementary Figure 6A-B), one-way ANOVA with post-hoc Dunnett multiple comparisons (Figure 2H-I, 3E, 4D, 7B-D, Supplementary Figure 7A-C) and unpaired Student’s t-test (Figure 1F-G, 1I-J, 1L-M, 4H, 5A-E, 6D-E, Supplementary Figure 1B, 4A, 4C, 11A-B). In all figures, p values are denoted as * < 0.05, ** < 0.01, *** < 0.001.

We added all of the controls in Figure 2B-C, 2G-H, 3B-C, 3G, 4C-E, 4I-J, 7B-D, and supplementary Figure 6A-B.

5. If possible, testing synthetic ascarosides would significantly improve the paper, though we understand that this may be difficult or impossible to achieve.

We understand that successful identification of the specific ascarosides will significantly improve the paper. However, our current data only support that the b-hydroxylated medium-chain ascarosides, such as the C13, C14, and C15, are the potential modulator pheromones. One of them may play a significant role, or they could synergistically modulate hermaphrodites NMJ. Considering this possibility, together with the consideration of time and cost, we did not synthesize and test these b-hydroxylated mediumchain ascarosides. We hope the reviewer could agree with us, and we will try to work on it in our future paper.

Reviewer #1:This interesting manuscript from Qian reports that *C. elegans* males may sex-specifically release compounds that alter NMJ physiology and locomotion in the hermaphrodite. Using a powerful combination of electrophysiology, behavioral analysis, optogenetics, genetic ablations, and imaging, the authors provide evidence that media conditioned by *C. elegans* males increases the E/I balance. Their results suggest that exposure during a critical period in late larval development is required for this effect; that exposure to conditioned media increases mEPSC frequency at the NMJ; that the effect is likely due to ascaroside pheromones; that these pheromones may be detected by the AWB neurons; that exposure to conditioned media changes pre- and post-synaptic CaV2 and AChR abundance, respectively; and that the CaV2 channel unc-2 is required for the physiological and behavioral effects of conditioned medium. A number of important issues remain unclear, such as the molecular identity of the relevant pheromone(s), whether these molecules are directly sensed by AWB, and whether changes at the NMJ are direct results of pheromone detection or are instead indirect changes that occur secondary to other changes in physiology that male pheromones can elicit in hermaphrodites. Nevertheless, the paper reports interesting and surprising findings that provide new insight into social and sexual interactions in *C. elegans* and the roles of pheromones in modulating physiology and behavior.1. A very significant concern (perhaps a fatal flaw) is that essential information about the methods and details of the experiments are lacking. Without these, it becomes impossible to assess the validity of the experimental design and interpretation. In particular, it is unclear how conditioned media (CM) was prepared and what method was used to expose animals to it. If the liquid culture method on p. 29 was used to prepare CM, this likely makes much of the data in the paper uninterpretable. Because these cultures are grown for 7 days, they likely differ greatly in the number of animals as well as their population structure (relative numbers of different larval stages, dauers, and adults of various ages). These cultures are also susceptible to bacterial and fungal contamination and cause animals significant stress. As such, direct comparisons of the activity of CM from N2, him-5, and daf-22 CM is highly problematic, as any of the above could account for differences in its activity. Without knowing that CM was prepared from a defined number of animals of defined ages, it's impossible to draw conclusions about the basis for the differential effects of CM. The absence of any experiments with synthetic ascarosides compounds these concerns.

We apologize for not including the information in the previous manuscript. We have added the detailed description of the conditioned media preparation in Figure 1C and Experimental model and subject details.

2. Statistics and data presentation. Not enough information is provided to understand how aldicarb sensitivity was statistically evaluated (it's particularly hard to understand the apparent significant differences in the results shown in 4C). Further, scatter plots need to be provided in place of the histograms shown in Figures1, 2, 5, 6, 7 and supplementary.

We included all of the detailed information of statistical analyses in page 34 line 14-22 of the manuscript:

“All data were reported as mean ± SEM (standard error of mean). Statistical analyses were performed using GraphPad Prism (version 7.0a). We calculated p values by two-way ANOVA with post-hoc Sidak multiple comparisons (Figure 1B, 1D, 2D-F, Supplementary Figure 2A-F, 4, 6A-B, 9A-G, 10A-C), one-way ANOVA with post-hoc Dunnett multiple comparisons (Figure 2B-C, 2G-I, 3B-C, 3E-G, 4C-E, 4I-J, 7B-C, Supplementary Figure 5A-B, 7A-C, ) and unpaired Student’s t-test (Figure 1F-G, 1I-J, 1L-M, 2B, 4H, 5A-E, 6DE, Supplementary Figure 1B, 3A, 3C 11A-B). In all figures, p values are denoted as * < 0.05, ** < 0.01, *** < 0.001.”

We also revised the figure legends accordingly.

Besides that, we replaced all of the histograms in Figure 1, 2, 5, 6, 7 and supplementary Figure 4, 11 with scatter plots.

Reviewer #2:The authors identify a fascinating and novel phenotype in hermaphrodite *C. elegans*, where synaptic transmission at the neuromuscular junction (measured using aldicarb response) and locomotion behavior are altered by the presence of male *C. elegans*. The authors demonstrate that this change is due to pheromones released from males, and that exposure to males during development is temporally dependent. The authors also do a preliminary dissection of what pheromones mediate the phenotype, the sensory circuitry required for this phenotype, and some mechanistic analysis at the NMJ. While the work is appropriate in scope and would be exciting for the readership of eLife, a number of the major conclusions of the manuscript are hard to evaluate due to lack of important experimental and statistical methodology and a number of crucial experiments lacking rigorous controls.1. Detailed descriptions of experimental methodology and statistical analyses are lacking for some of the most crucial experiments and thus a number of the major findings are not able to be evaluated in a rigorous manner.Aldicarb assays – specifics on aldicarb plate is needed- "Briefly,1 mM aldicarb was added to the NGM plate". Variability between aldicarb plates is common and often mentioned in *C. elegans* literature, thus specific details are needed to evaluate how plates were made and maintained, and controls need to be included for experiments in order to evaluate and compare results. Specific details on what values were compared using t-test for aldicarb assays is unclear and needs to be described (eg. on a single time point?).Conditioned media experiments – This reviewer could find no method descriptions for conditioned media experiments- How was male and hermaphrodite conditioned media made and administered to worms? This is critical for many of the experiments and is especially important to know for male-conditioned media generated from non-him mutant strains (daf-22, N2, and TR389).

Thanks for the reviewer’s suggestion.

1) We included detailed description of aldicarb assay and conditioned medium preparation in Experimental model and subject details.

“Aldicarb assay

The aldicarb assay was performed as previously described (Vashlishan et al., 2008). Aldicarb (ApexBio) was dissolved in ethyl alcohol and added to NGM at a final concentration of 1.4mM (Testing hermaphrodites) or 0.5 mM (Testing males). These plates (35mm) were seeded with 75 ul OP50 and allowed to dry overnight before use. More than 20 animals at the young adult stage (otherwise indicated) were picked on an aldicarb plate for aldicarb assay. Animals were scored as paralyzed when they did not respond to the platinum wire prodding. The paralyzed animals were counted every 10 or 15 minutes. At least three double-blind replicates for each group were tested.”

“Preparation of conditioned media

Hermaphrodite and male-secreted metabolites were collected according to a previous publication (Srinivasan et al., 2008). Synchronized *C. elegans* (WT [N2], *him-5* [N2], WT [TR389], *him-5* [TR389], *daf22* [N2], and *daf-22*; *him-5* [N2]) with a density of 10,000 worms/plates (90 mm) were grown on the nematode growth media (NGM) agarose (seeded with *E. coli* strain OP50) at 20 °C. There were 43.07 ± 0.77%, 37.29 ± 1.28%, and 39.26 ± 1.55% males in *him-5* (N2), *him-5* (TR389), and *daf-22*; *him-5* (N2) strains respectively. After worms reached the young adult stages, they were collected by settling for 10 minutes and were washed three times with M9 buffer to remove bacteria. To further remove the bacteria in the gut, the worms were then placed in M9 buffer in a shaker (150 rpm) at 20 °C for 30 minutes, and rinsed three times with ddH_2_O. Subsequently, worm-secreted metabolites were collected by incubating the worms in ddH_2_O in a shaker (150 rpm) for 3 hours with a density of 30,000 worms/ml. Afterward, the worms were removed by settling on ice for 5 minutes. The metabolites were filtered through 0.22 μm filters, aliquoted, and stored at -80 °C. For conditioned medium preparation, 10 μl metabolites mixed with 90 μl OP50 *E. coli* was spread on a 35 mm NGM plate. Plates were allowed to dry overnight before use.”

2) For aldicarb assay analysis, two-way ANOVA with post-hoc Sidak multiple comparisons was used to compare the differences between groups.

2. Appropriate controls are missing in many experiments or are shown in separate graphs making it hard for reviewers/readers to assess phenotypes compared to positive and negative controls, or to evaluate if mutant or transgenic worms have phenotypes in control conditions. Examples include Figure 2D-F, Figure 3C, Figure 4C-L, Figure 7A-E

We have added appropriate controls in Figure 2B-C, Figure 2G-H, Figure 3B-C, Figure 3G, Figure 4C-E, Figure 4I-J, and Figure 7A-D.

For Figure 2D-F, these experiments were simultaneously performed, and they could be controlled for each other.

3. The results in Figure 2 demonstrate temporal dependence, however, to conclude that L4 is the important stage ("exposure of the male environment in the entire L4 stage is critical for modulation of the NMJ synaptic transmission in hermaphrodites" "male environment exposure at a critical period (the L4 stage) is required for the modulation of hermaphrodites NMJ cholinergic synaptic transmission.") would require exposure to male-conditioned media during only the L4 stage, as well as after larval development (adult only exposure to male-conditioned media). Some effects shown here could be explained by duration of exposure. An ANOVA should be performed to allow readers to compare between all timepoints for both herm and male conditioned media conditions.

We added two additional temporal conditions in Figure 2: L4 out (get out of male-conditioned medium at L4 stage) and adult out (get out of the male-conditioned medium at the adult stage) (Figure 2A, 2C, and supplementary Figure 2E). The data shows that “adult out” hermaphrodites present the increased aldicarb sensitivity comparable to those sustained in the male excretome environment, but not the “L4-out” hermaphrodites. Together with the data in Figure 2B, we proved that the L3-L4 is an important stage for male pheromone modulation.

Besides, in Figure 4E, we showed that optogenetically activating AWB neurons only in L4 is sufficient to modulate NMJ synaptic transmission. Collectively, the data support the notion that L4 is a critical stage for modulating the NMJ synaptic transmission. We revised the manuscript in page 8 line 21-28 and page 9 line 1-8.

4. For the figure and section "Excitatory postsynaptic receptor clustering is increased in hermaphrodites exposed to the male environment" – fluorescence intensity is not a good measure of synaptic structure or clustering. Additional markers beyond rab-3 (e.g. cla-1) are needed to evaluate presynaptic morphology and ideally pre and post synaptic clustering experiments would include quantification of the number or size of synaptic puncta.

According to the suggestion, we measured another presynaptic protein, UNC-57/endophilin, to evaluate presynaptic morphology. The fluorescence intensities and puncta densities of UNC-57/endophilin and RAB-3 at excitatory and inhibitory synapses are unaltered, suggesting that neither excitatory nor inhibitory synapse structures are affected by the modulator ascarosides. We included the data in Supplementary Figure 11.

We also measured the postsynaptic receptor (ACR-16, UNC-29, and UNC-49) puncta densities, and none of them show differences in hermaphrodites under hermaphrodite- and male-conditioned medium (Figure 5C-D).

5. The analysis of ascarosides in male vs herm conditioned media identifies a subset of "modulator ascarosides" and the genetic experiments suggest these may be responsible for the phenotype of male conditioned media. However, without testing the direct effect of the ascarosides on the phenotype, the wording of subsequent results and experiments throughout needs to be toned down ("Thus, the modulator ascarosides cause increased postsynaptic receptor clustering at excitatory synapses in hermaphrodites.", "These findings collectively indicate that the modulator ascarosides specifically promote accumulation of CaV2 calcium channel at excitatory cholinergic synapses, and the increased CaV2 calcium channel abundance account for the potentiated cholinergic synaptic transmission at NMJ.", etc). This is mentioned by the authors in discussion, but needs to be applied to the Results sections ("Although we provided extensive genetic evidence, we have not experimentally confirmed that these specific ascarosides are sufficient to modulate hermaphrodite NMJ synaptic transmission.").

Thanks for the reviewer’s suggestion, and we revised the Results in page 18 line 3-5 and page 19 line 20-23.

Reviewer #3:In this manuscript, Qian et al., describes mechanisms how *C. elegans* male pheromones mediate sexually dimorphic impact on synaptic transmission at the NMJ and thus locomotive behaviors. First, the authors found that the male environment, which represents a male-conditioned medium, increases NMJ excitation and inhibition balance of hermaphrodites by specifically increasing cholinergic transmission but not GABAergic transmission, resulting in decreased locomotive activities in hermaphrodites. Then they identified a critical period (the L4 larval stage) required for modulation of the male-conditioned medium on NMJ synaptic transmission in hermaphrodites. Second, by performing candidate mutant test and LC-MS analysis, they identified chemical components in the male-conditioned medium, which are the medium-chain β-hydroxylated ascaroside pheromones. These pheromones appeared to be sensed by the AWB chemosensory neurons and transmitted via cGMP signaling. Finally, they showed that hermaphrodites exposed to the male-conditioned medium exhibit increased postsynaptic nicotinic acetylcholine receptor clustering as well as increased presynaptic CaV2 calcium channel localization at NMJ.Overall, I think this is a very interesting story, with novel findings and implications about the pheromone-mediated behavioral plasticity in this important model organism. There are a couple of places where I think the authors could fill gaps in the story that would make it more complete and logically compelling.Several major points that need to addressed and improved include;1. Authors should test whether chemically synthesized ascarosides, including at least C13 and C14, individually and in a mixture, recapitulate effects of the male-conditioned medium on aldicarb assay as well as electrophysiology experiment. Without these data, it is hard to assure that these pheromones mediate it.

We understand that successful identification of the specific ascarosides will significantly improve the paper. However, our current data only support that the b-hydroxylated medium-chain ascarosides, such as the C13, C14, and C15, are the potential modulator pheromones. One of them may play a significant role, or they could synergistically modulate hermaphrodites NMJ. Considering this possibility, together with the consideration of time and cost, we did not synthesize and test these b-hydroxylated mediumchain ascarosides. We hope the reviewer could agree with us, and we will try to work on it in our future paper.

We revised our results and conclusion to a weaker statement. We hope that the reviewer could agree with us, and we will try to work on it in our future paper.

2. Authors claimed that the AWB chemosensory neurons detect these pheromones and mediate effects of male-conditioned medium mainly by performing cell-ablation and mutant studies. Nowadays, it is quite common to do Ca^2+^ imaging experiments using genetically encoded Ca^2+^ sensors, which make a solid conclusion that AWB neurons detect these pheromones.

We have conducted calcium imaging to monitor intracellular Ca^2+^ dynamics upon male pheromone stimulation in AWB soma. We found that the AWB neurons elicited a rapid and robust calcium transient response to the male excretomes, but not hermaphrodite excretomes. We included this data in Figure 4F-H and revised our manuscript in page 16 line 3-8.

3. Authors have not addressed or discussed at all why male affects hermaphrodite locomotion. Is mating efficiency or physiology such as egg-laying of hermaphrodites altered in the male environment?

Thanks for the reviewer’s suggestion. We added discussion about the locomotion changes by male pheromone in page 10 line 14-16.

Besides, we compared the mating efficiency and brood size of hermaphrodites from hermaphrodite- and male-conditioned medium. We observed that the mating efficiency is increased in hermaphrodites from the male-conditioned medium, but there are no significant differences in brood size between the two groups. We added the data in Figure 2I and Supplementary Figure 5.

[Editors’ note: what follows is the authors’ response to the second round of review.]

The reviewers feel that your manuscript has been significantly improved by the inclusion of new data and clarification of some of technical points. However, there are several additional points that need to be addressed before acceptance, as outlined by Reviewer #2. In particular, please be sure to address each of the four "major comments" noted by Reviewer #2 in your resubmission. None of these require new experiments, but instead can be addressed by carrying out additional statistical tests and/or editing the text.Reviewer #2:This revised manuscript is very exciting, and will be of interest to the readership of eLife. The authors have included many revisions and new experiments that make the revised manuscript rise to the level of publication in eLife in terms of rigor and novelty. There are a few changes that would further improve clarity on some experiments.1. While the authors have included much more detail on statistical analysis and methods, it should be clearly stated exactly what comparison the p-values represent for aldicarb assays (eg. Figure 1b- all time points? One timepoint?).

According to the reviewer’s suggestion, we revised the figure legends in Figure 1B, 1D, 2D-F, Figure 2—figure supplement 1A-E, Figure 2—figure supplement 2, Figure 2—figure supplement 4, Figure 4—figure supplement 2A-G, and Figure 4—figure supplement 3A-C by stating “two-way ANOVA comparing all of the time points”.

2. Figure 2C. Please provide p-value comparing L4 out vs adult out

We revised Figure 2C by adding the p-value to compare L4 out vs. adult out.

3. The experiment in figure 4E appears to show the addition of ATR lowers aldicarb sensitivity and that light exposure rescues this. P-values comparing all conditions should be included. While ATR can impact behavior and does not necessarily take away from the finding, the wording should be altered to reflect if there are differences between the non-ATR controls and the ATR and ATR + light condition.Pg 15 line 21 "Hermaphrodites with activated AWB neurons during the L4 stage showed higher sensitivity to aldicarb than controls without blue light activation (Figure 4E)."This statement suggests that the final bar is higher than all control bars, which is not shown statistically.

Yes, the addition of ATR lowers aldicarb sensitivity for unknown reasons (ATR+ light- vs. ATRlight-), which may cause no increase of aldicarb sensitivity in ATR+ light+ comparing with ATRlight+. We revised our manuscript to include the ATR effects on aldicarb sensitivity and clarify the “controls” on page 16, line 405, and added p-value in Figure 4E.

“We observed decreased aldicarb sensitivity in animals fed with ATR (ATR+ light- vs. ATRlight-). Nevertheless, hermaphrodites with activated AWB neurons during the L4 stage showed higher sensitivity to aldicarb than controls without blue light activation (ATR+ light+ vs. ATR+ light-) (Figure 4E). This effect is absent in the groups lacking ATR (ATR- light+ vs. ATR- light-) (Figure 4E).”

4. In Figure 4D, it appears that the velocity of worms with AWB ablation is reduced in both the herm- and male-cond compared with no light controls. Please provide p-values comparing all conditions, and if the herm-cond AWB ablation is reduced compared with no light herm-cond, please alter the wording of the statement below to reflect this data."Pg 15 line 14. Our data showed that AWB neurons ablation blocked the decreased velocity by the modulator ascarosides (Figure 4D)."

We added p-values in Figure 4D and revised the manuscript on page 15, line 397.

“Our data showed that ablation of AWB neurons decreased the locomotion velocity in hermaphrodites from the hermaphrodite-conditioned medium. In addition, ablation of AWB neurons blocked the decreased velocity by the modulator ascarosides (Figure 4D).”